# Conformational plasticity of *RAS Q61* family of neoepitopes results in distinct features for targeted recognition

Andrew C. McShan [1,2,4,7], David Flores-Solis [3,5,7], Yi Sun [1,2], Samuel E. Garfinkle [1,2], Jugmohit S. Toor [3,6], Michael C. Young [1] & Nikolaos G. Sgourakis [1,2] ✉

The conformational landscapes of peptide/human leucocyte antigen (pHLA) protein complexes encompassing tumor neoantigens provide a rationale for target selection towards autologous T cell, vaccine, and antibody-based therapeutic modalities. Here, using complementary biophysical and computational methods, we characterize recurrent $RAS_{55-64}$ Q61 neoepitopes presented by the common HLA-A*01:01 allotype. We integrate sparse NMR restraints with Rosetta docking to determine the solution structure of $NRAS^{Q61K}$/HLA-A*01:01, which enables modeling of other common $RAS_{55-64}$ neoepitopes. Hydrogen/deuterium exchange mass spectrometry experiments alongside molecular dynamics simulations reveal differences in solvent accessibility and conformational plasticity across a panel of common Q61 neoepitopes that are relevant for recognition by immunoreceptors. Finally, we predict binding and provide structural models of $NRAS^{Q61K}$ antigens spanning the entire HLA allelic landscape, together with in vitro validation for HLA-A*01:191, HLA-B*15:01, and HLA-C*08:02. Our work provides a basis to delineate the solution surface features and immunogenicity of clinically relevant neoepitope/HLA targets for cancer therapy.

The display of epitopic peptides by class I major histocompatibility complex (MHC-I) proteins to CD8[+] cytotoxic T lymphocytes (T cells) forms the cornerstone of adaptive immunity, enabling surveillance of the endogenous proteome against invading pathogens and intracellular oncoproteins[1]. Properly conformed peptide/MHC-I complexes (pMHC-I) are assembled in the endoplasmic reticulum, where a multisubunit peptide loading complex coordinates the association of transported peptides with heavy and light chain components[1–3]. While the light chain (human $\beta_2$-microglobulin, h$\beta_2$m) is invariant across each pMHC-I, the heavy chain is encoded by the most polymorphic loci

in the human genome, with thousands of identified allotypes categorized in the canonical human leukocyte antigen (HLA) A-, B-, and C-classes[4]. The HLA peptide binding groove comprises polymorphic residues distributed among six distinct pockets (termed A through F) that define the repertoire of epitopic peptides displayed by each HLA[5]. Peptide binding specificity is determined by anchoring interactions with HLA side chains forming the A-F-pockets, which impose chemical and steric restrictions on peptide docking[6]. Thus, the conformational landscape of pMHC-I structures is defined by both the peptide sequence and HLA groove polymorphisms, which has important

[1]Center for Computational and Genomic Medicine, Department of Pathology and Laboratory Medicine, The Children's Hospital of Philadelphia, Philadelphia, PA 19104, USA. [2]Department of Biochemistry and Biophysics, Perelman School of Medicine, University of Pennsylvania, Philadelphia, PA 19104, USA. [3]Department of Chemistry and Biochemistry, University of California, Santa Cruz, CA 95064, USA. [4]Present address: School of Chemistry & Biochemistry, Georgia Institute of Technology, 901 Atlantic Dr NW, Atlanta, GA 30318, USA. [5]Present address: German Center for Neurodegenerative Diseases (DZNE), Von-Siebold Straße 3A, 37075 Göttingen, Germany. [6]Present address: Immunology Research Program, Henry Ford Cancer Institute, Henry Ford Health, Detroit, MI 48202, USA. [7]These authors contributed equally: Andrew C. McShan, David Flores-Solis. ✉e-mail: nikolaos.sgourakis@pennmedicine.upenn.edu

ramifications for recognition by antibodies, T-cell receptors, and NK cell receptors[7–11]. Since the surface chemistry of pMHC-I proteins dictates recognition by T-cell receptors (TCRs) and therapeutic monoclonal antibodies[12–14], structure determination of disease-associated complexes is essential for understanding immune responses, and the development of targeted therapies.

In recent years, there has been significant progress in the identification of immunogenic, mutated self-peptides (termed neoepitopes) presented on the surface of cancerous cells by HLA molecules[15,16]. Functional and structural characterization of neoepitope/HLA complexes enables design of targeted immunotherapies against cancer, such as engineered TCRs, antibodies, and cancer vaccines[17–19]. In particular, mutations in the RAS family of GTPase genes (NRAS, KRAS, and HRAS) are found in approximately 25% of patients with neuroblastoma and melanoma, where mutation at consensus residues 12, 13, and 61 are associated with driving oncogenic activity[20,21]. While structures of HLAs presenting $RAS_{5-14}$ neoepitopes derived from mutations at residues 12 and 13 have been solved[22,23], pMHC-I structures derived from $RAS_{55-64}$ residue 61 neoepitopes have remained elusive. Using an integrated workflow to analyze tumor sequencing data from neuroblastoma patients, ProTECT[24], we previously predicted that a public, decameric epitope derived from the recurrent $NRAS_{55-64}$ mutation, Q61K (ILDTAGKEEY), could interact with the common HLA-A*01:01 molecule, in agreement with experimental validation of binding in vitro and in vivo by other groups[25–27]. More than 60% of NRAS positive tumors harbor Q61 mutants, making this epitope an attractive target for cancer therapy[21,28]. In addition, targeting of $NRAS^{Q61K}$ by T cells has shown great promise in inhibiting tumor growth and angiogenesis in a peptide/HLA-restricted manner[27,29,30]. Notwithstanding, the structure of the $NRAS^{Q61K}$/HLA-A*01:01 complex has not yet been solved, limiting insights into how different $RAS_{55-64}$ Q61 mutations could influence recognition by TCRs and antibody engineering approaches. While pMHC-I studies by X-ray crystallography have provided hundreds of structures in the Protein Data Bank[9,31], many important pMHC-I complexes are not amenable to X-ray diffraction studies, despite exhaustive screening efforts[32]. Computational approaches have enabled modeling of pMHC-I structures using de novo or homology-based approaches[33–35]. However, both the precision and accuracy of these methods are limited by the large conformational space, which can be adopted by different peptide sequences within the HLA groove[33].

To address these bottlenecks, we used complementary biophysical techniques to elucidate the assembly, structure, and conformational plasticity of peptide: HLA complexes encompassing different NRAS epitopes in a physiologically relevant, aqueous environment. Focusing on the $NRAS^{Q61K}$/HLA-A*01:01/h$\beta_2$m complex, we demonstrate that empty (peptide-free) HLA molecules undergo a conformational transition to sample a peptide-receptive conformation, which can spontaneously associate with NRAS peptides. We further implemented methyl-selective isotopic labeling for recording intermolecular NMR restraints, and combined our NMR data with Rosetta flexible docking simulations to determine the solution structure of the pMHC-I complex. We then utilized molecular dynamics (MD) simulations alongside hydrogen-deuterium exchange mass spectrometry (HDX-MS) to probe the solvation and conformational dynamics of sites within the MHC groove of $NRAS^{Q61K}$, as well as wild-type (Q61) and three additional, common neoepitopes (Q61H, Q61L, Q61R). Our data reveal that single amino acid changes at residue 61 of NRAS alter the peptide conformational landscape, where Q61K and Q61L display the greatest and least amount of mobility, respectively. Finally, we examined the HLA binding repertoire of NRAS neoepitopes using both in silico and in vitro approaches. These results underscore the public presentation of $RAS_{55-64}$ Q61 neoepitopes beyond HLA-A*01:01, including the HLA-A*01:191, HLA-B*15:01, and HLA-C*08:02 allotypes. Taken together, our results highlight differences in conformational plasticity, solvation patterns, and surface accessibility across NRAS/HLA protein complexes, which can be exploited to design neoepitope-selective immunoreceptors and other therapeutic modalities.

## Results

### $NRAS^{Q61K}$ binds to the common HLA-A*01:01 allotype with a moderate affinity

The $RAS_{55-64}$ epitope sequence is strongly conserved across the RAS protein family (Supplementary Fig. 1). While we chose the $NRAS_{55-64}$ Q61K peptide as a model system, Q61 mutations in $KRAS_{55-64}$, $NRAS_{55-64}$, and $HRAS_{55-64}$ result in identical neoepitope sequences. Thus, $NRAS_{55-64}$ Q61K is a representative for the entire family of $RAS_{55-64}$ Q61K neoepitopes. Towards biophysical and structural characterization, we recombinantly expressed HLA-A*01:01 heavy chain and h$\beta_2$m light chain proteins in E. coli and performed in vitro refolding together with a 10-fold molar excess of the $NRAS^{Q61K}$ peptide[36]. Purification by size exclusion chromatography (SEC) revealed properly folded, monomeric pMHC-I complexes (Supplementary Fig. 2a). Liquid chromatography–mass spectrometry (LC–MS) confirmed the presence of the bound $NRAS^{Q61K}$ peptide with an observed mass of 1138 Da (Supplementary Fig. 2b). Differential scanning fluorimetry (DSF) revealed a moderately stable complex with melting temperature ($T_m$) of 53 °C (Supplementary Fig. 2c), consistent with reports of pMHC-I complexes exhibiting $T_m$'s ranging from 41 to 67 °C[37,38,39]. We next carried out fluorescence anisotropy competition experiments to estimate the affinity of $NRAS^{Q61K}$ for the HLA-A*01:01/h$\beta_2$m complex[40]. Incubation of TAMRA-dye labeled $NRAS^{Q61K}$ with $NRAS^{Q61K}$/HLA-A*01:01/h$\beta_2$m and varying concentrations of unlabeled competitor $NRAS^{Q61K}$ yielded an $IC_{50}$ of 260 nM (Supplementary Fig. 2d), consistent with a predicted $IC_{50}$ of 218.5 nM by netMHCpan-4.1[41]. We also attempted to solve the structure of the $NRAS^{Q61K}$/HLA-A*01:01/h$\beta_2$m complex by X-ray crystallography. However, following multiple crystallization screens, we were unable to find conditions that allowed crystal formation. Together, these results suggest that, while $NRAS^{Q61K}$ can associate with the HLA-A*01:01 groove, the resulting complex is moderately stable leading to high levels of background exchange with the empty form, which likely prohibits the formation of an ordered crystal lattice for structural studies by X-ray diffraction.

### Assembly of $NRAS^{Q61K}$ with empty HLA-A*01:01 induces global structural changes

We sought to study the $NRAS^{Q61K}$/HLA-A*01:01/h$\beta_2$m assembly mechanism and structure using solution NMR, where structures of proteins up to 82 kDa have been determined de novo[42]. The size and complexity of our system (a non-symmetrical 45 kDa heterotrimer) necessitated the use of selective isotope labeling schemes (Supplementary Fig. 3a–e, Supplementary Table 1). Here, we applied complementary labeling schemes, inspired by the amino acid composition of the HLA-A*01:01 groove where methyl-bearing (Ala, Ile, Leu, Val) and aromatic (Phe, Tyr) residues lining the A- to F-pockets which anchor the peptide (Supplementary Fig. 3b). Our selective labeling allowed us to obtain unambiguous chemical shift assignments of resonances corresponding to both the HLA-A*01:01 heavy chain and $NRAS^{Q61K}$ peptide residues, as well as to record intra- and intermolecular NOEs between HLA-A*01:01 and $NRAS^{Q61K}$ protons. Chemical shift assignments of HLA-A*01:01 were established using TROSY-based 3D HNCA, 3D HN(CA)CB, and 3D HNCO triple-resonance together with 3D NOESY experiments, as shown for multiple other pMHC-I systems in previous studies by our group[8,43,44–48,49]. Free $NRAS^{Q61K}$ peptide chemical shift assignments were performed using 2D $^1H$-$^1H$ NOESY and TOCSY experiments[50]. Bound peptide resonance assignments were achieved using transfers from the free state, followed by validation with peptide/HLA NOE cross-peaks obtained using samples with $^{15}N$/$^{13}C$ labeled $NRAS^{Q61K}$. Altogether, we achieved 100% stereospecific assignment of the 89 AILV methyl, 85% of the 21 FY aromatic as well as 93% of the

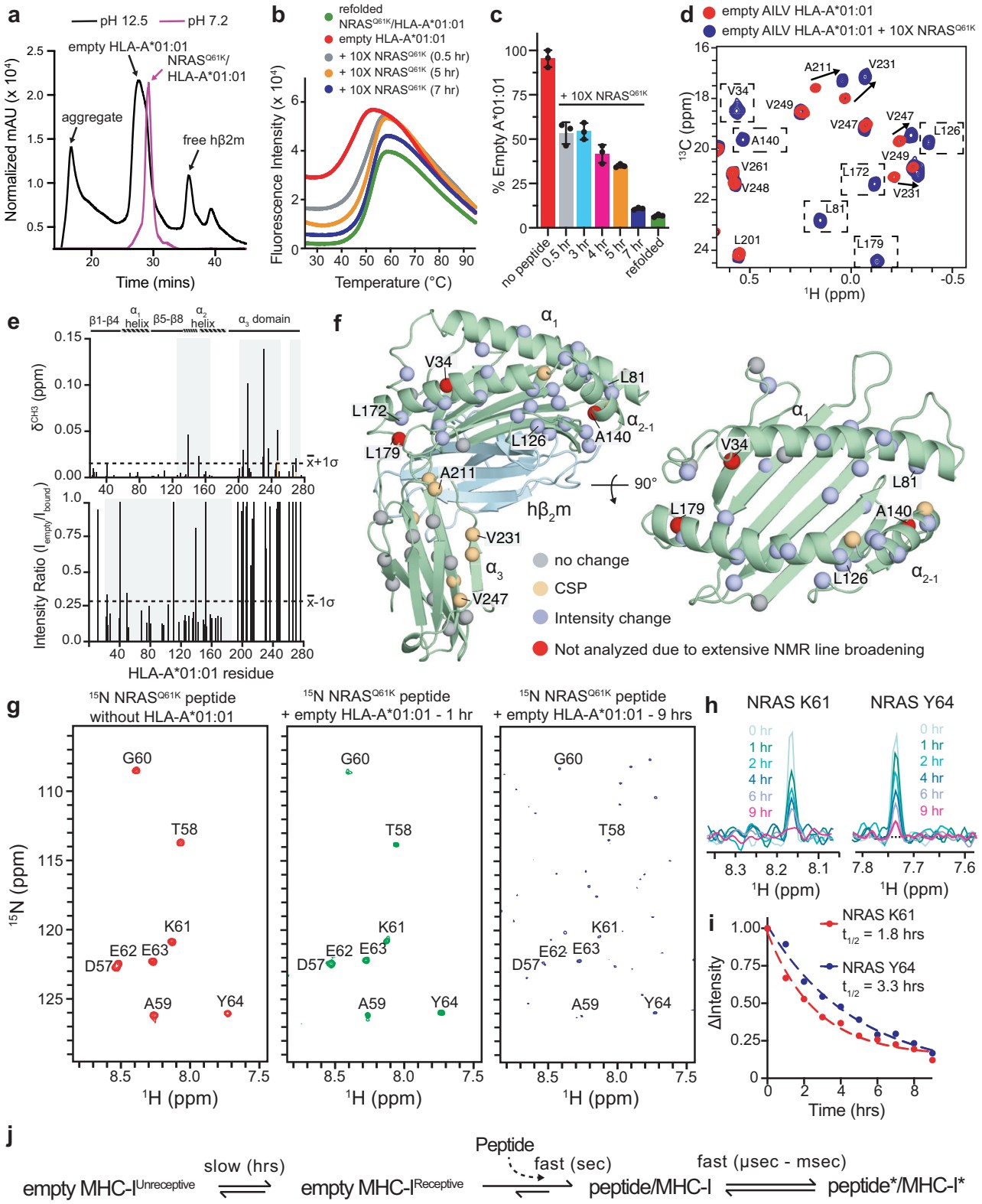

**g** | ¹⁵N NRAS^Q61K peptide without HLA-A*01:01 | ¹⁵N NRAS^Q61K peptide + empty HLA-A*01:01 - 1 hr | ¹⁵N NRAS^Q61K peptide + empty HLA-A*01:01 - 9 hrs

**h** NRAS K61 | NRAS Y64

**i** NRAS K61 t₁/₂ = 1.8 hrs | NRAS Y64 t₁/₂ = 3.3 hrs

**j** empty MHC-I^Unreceptive ⇌ [slow (hrs)] empty MHC-I^Receptive ⇌ [Peptide, fast (sec)] peptide/MHC-I ⇌ [fast (μsec – msec)] peptide*/MHC-I*

amide groups of HLA-A*01:01 (Supplementary Fig. 3c–e), and complete assignment of peptide atoms in proximity to the groove.

Direct peptide interactions with empty MHC-I molecules remain poorly understood but are thought to be a driving factor in antigen selection and pMHC-I cell surface stability. Thus, we sought to characterize how NRAS^Q61K could affect the structure and stability of empty MHC-I complex. In vitro studies of peptide interactions with empty MHC-I complexes have been previously reported in the literature, and

are often employed as a first approximation pMHC-I interactions in vivo[38,51–53]. To characterize how NRAS^Q61K assembles with HLA-A*01:01, we first emptied HLA-A*01:01/hβ₂m complexes of their in vitro refolded peptide by brief exposure to pH 12.5 followed by SEC purification (Fig. 1a), as previously described[54–56]. LC–MS confirmed that basic pH treatment of HLA-A*01:01/hβ₂m resulted in monomeric, empty molecules, while control treatment with pH 7.2 revealed the NRAS^Q61K peptide was still bound in the complex (Supplementary

**Fig. 1 | Conformational changes and assembly kinetics for NRAS$^{Q61K}$ association with empty HLA-A*01:01/hβ$_2$m. a** SEC traces of HLA-A*01:01/hβ$_2$m following short exposure to neutral (magenta) or basic (black) phosphate buffer. **b** DSF of 7 μM HLA-A*01:01/hβ$_2$m following short exposure to pH 12.5 phosphate buffer. Red curve – no peptide added. 10× excess NRAS$^{Q61K}$ peptide was added and incubated for 0.5 (gray), 5 (orange), and 7 (blue) hrs. Green curve – refolded NRAS$^{Q61K}$/HLA-A*01:01/hβ$_2$m control. **c** Percent empty HLA-A*01:01 determined by DSF upon incubating empty HLA-A*01:01 without or with 10× excess NRAS$^{Q61K}$ peptide for the indicated amount of time. Refolded complex is shown for reference. Data are mean ± SD for $n = 3$ technical replicates. **d** 2D $^1$H-$^{13}$C methyl HMQC spectra of 50 μM AILV labeled HLA-A*01:01 refolded with natural abundance hβ$_2$m following exposure to pH 12.5 phosphate buffer (red – no peptide; blue – 10× excess NRAS$^{Q61K}$ peptide added) recorded at 25 °C at a $^1$H field of 800 MHz. Dotted boxes represent methyl resonances exhibiting line broadening in the empty state. **e** Chemical shift perturbations (CSP, δ$^{CH3}$, ppm) (top) and intensity ratios (I$_{empty}$/I$_{bound}$) (bottom) for

AILV methyl probes of HLA-A*01:01 with 10× excess NRAS$^{Q61K}$ (I$_{bound}$) relative empty HLA-A*01:01 (I$_{empty}$). Dotted lines represent the average plus one standard deviation for CSP analysis or minus one standard deviation for intensity ratio analysis. Gray boxes highlighted affected regions. The protein domains of HLA-A*01:01 are shown for reference. **f** Mapping of HLA-A*01:01 methyl residues with resonances exhibiting either CSP or intensity changes upon binding to NRAS$^{Q61K}$ onto the X-ray structure of HLA-A*01:01 (PDB ID 6AT9, peptide atoms removed). **g** 2D $^1$H-$^{15}$N HMQC spectra of $^{15}$N labeled NRAS$^{Q61K}$ without (red) or with empty unlabeled HLA-A*01:01/hβ$_2$m after 1 h (green) or 9 h (blue) of incubation recorded at a $^1$H field of 800 MHz at 25 °C. **h** 1D $^1$H spectral slices for the amide of K61 and Y64 of $^{15}$N labeled NRAS$^{Q61K}$ throughout incubation in panel (**g**). **i** The change in NMR signal intensity (ΔIntensity) as a function of incubation time and fitted half-life (t$_{1/2}$) values for K61 and Y64. **j** Schematic of the timescale for different conformational changes in the MHC-I complex.

Fig. 4a, b). Empty HLA-A*01:01/hβ$_2$m complexes were receptive for binding to NRAS$^{Q61K}$ as revealed by both DSF and solution NMR titrations (Fig. 1b–d). To determine the kinetics of peptide binding to the HLA, DSF experiments were performed where purified, empty HLA-A*01:01 was offered 10-fold molar excess of NRAS$^{Q61K}$ peptide, and incubated for periods ranging from 0.5 to 7 h (Fig. 1b, c). The addition of NRAS$^{Q61K}$ resulted in a reduction in the population of empty HLA-A*01:01 by approximately 50% within the first 0.5 h of incubation (Fig. 1b, c). Interestingly, complete reduction in the population of empty HLA-A*01:01 to levels observed in control experiments using refolded NRAS$^{Q61K}$/HLA-A*01:01 required incubation with NRAS$^{Q61K}$ for up to 7 h (Fig. 1b, c). These results suggest the presence of a population of empty, peptide-receptive HLA-A*01:01 molecules, which can associate spontaneously with NRAS$^{Q61K}$, as well as a population of species undergoing a slow conformational change (several hours) towards the peptide-receptive state.

We also performed NMR titrations to uncover structural differences in the empty versus peptide bound HLA-A*01:01 groove. Compared to the NRAS$^{Q61K}$ bound state, 2D $^1$H-$^{13}$C methyl HMQC spectra of AILV labeled empty HLA-A*01:01 exhibited both chemical shift perturbations (CSPs) and exchange-induced line broadening of NMR resonances corresponding to methyl groups spanning the entire HLA groove, including sites on the α$_1$ helix, α$_2$ helix, and pleated β-sheet (Fig. 1d–f, Supplementary Fig. 5a, b). These findings are in agreement with previous NMR studies of empty MHC-I molecules[49,56,57]. Intensity changes were primarily observed for NMR resonances corresponding to methyl groups located in the HLA-A*01:01 groove, suggesting the presence of dynamic motions on the intermediate (microseconds to milliseconds) timescale in the absence of bound peptide (Fig. 1e, f). The observation of CSPs at the interface between the HLA-A*01:01 α$_3$ domain and hβ$_2$m, distal to the peptide binding groove, revealed a rearrangement of the MHC-I domain structure (Fig. 1e, f). Together, these data support a global conformational change in the HLA-A*01:01/hβ$_2$m complex, induced by peptide binding.

To follow peptide binding kinetics, we obtained U-[$^{15}$N] labeled NRAS$^{Q61K}$ and performed a series of incubations with empty HLA-A*01:01/hβ$_2$m complex, prepared at natural isotopic abundance. We monitored peptide/MHC-I association as a function of time using 2D $^1$H-$^{15}$N HMQC spectra. Upon binding to HLA-A*01:01/hβ$_2$m, NMR resonances corresponding to the NRAS$^{Q61K}$ amide groups exhibited increased line broadening, likely due to increased transverse relaxation rate of the peptide $^{15}$N and $^1$H atoms within the much larger (45 kDa) pMHC-I complex (Fig. 1g, h). The change in NRAS$^{Q61K}$ amide NMR signal intensity was followed as a function of time and fit to a single-phase exponential decay model, which revealed half-lives of peptide association with empty HLA-A*01:01/hβ$_2$m complex of up to 3.7 h (Fig. 1i, Supplementary Fig. 6). Taken together with previous studies[58–60], our DSF and NMR results revealed that empty MHC-I molecules undergo a conformational change to a peptide-receptive

state, which occurs on a slow (hours) timescale (Fig. 1j). Once a receptive state is formed, peptide association with the MHC-I proceeds via a fast (msec-sec timescale) process (Fig. 1j). Following pMHC-I complex formation, further conformational changes can be observed for both the bound peptide and the HLA groove on a fast (μsec - msec) timescale (Fig. 1j), in agreement with previous studies[7,8,56,61,62].

## The NRAS$^{Q61K}$/HLA-A*01:01 NMR structure reveals key features for targeted recognition

Towards determining the structure of the NRAS$^{Q61K}$/HLA-A*01:01 complex, we collected distance restraints in the form of intra- and intermolecular NOEs (Fig. 2a–c, Supplementary Fig. 3a, Supplementary Table 2, Supplementary Table 3). We recorded a suite of 3D SOFAST NOESY-HMQC experiments to obtain NOE cross-peaks at high signal-to-noise and spectral resolution[63]. These experiments allowed for simultaneous detection and assignment of intramolecular NOEs (between HLA-A*01:01 residues) and intermolecular NOEs (between HLA-A*01:01 and NRAS$^{Q61K}$ residues) spanning the entire HLA groove (Fig. 2a–c). NOE cross-peaks recorded using 3D SOFAST NOESY-HMQC experiments were also observed in an analogous isotope-filtered/edited pulse sequences performed under independent conditions, albeit with lower sensitivity (Supplementary Fig. 7a–e)[64].

We then used both intra- and intermolecular NOE-derived constraints to perform structure calculations of the NRAS$^{Q61K}$/HLA-A*01:01/hβ$_2$m complex with the FlexPepDock ab-initio protocol[65] (Fig. 3a). FlexPepDock was chosen as previous benchmarks showed that even in the absence of experimental constraints, the protocol can sample near native pMHC-I conformations[64,66]. As a starting input template, we used a previously solved high-resolution structure of HLA-A*01:01 in complex with a different decameric peptide (PDB ID 6AT9) with the atoms corresponding to the original peptide removed[26]. Following refinement of the HLA-A*01:01 heavy chain using intramolecular NOEs, an atomic model of the free extended NRAS$^{Q61K}$ peptide was placed near the HLA-A*01:01 groove in an orientation where P2 and P9 anchor positions of the peptide were oriented towards the A- and F-pockets of the MHC-I, respectively. This orientation is based on our observed peptide/HLA NOEs (Fig. 2a), which corroborates known peptide binding motifs associated with the HLA-A*01:01 groove, including canonical position 2 anchors (preferred ST, tolerated AILMV) and C-terminal anchors (preferred Y, tolerated F)[67]. A total of 50,000 models were generated from the initial starting structure, followed by fast low-resolution modeling and refinement of low-resolution models with and without intra- and intermolecular NOE-derived NMR constraints (Fig. 3b–e). We clustered the resulting structures, sorted them according to the total Rosetta energy score, and filtered out structures with NOE violations. Following the FlexPepDock ab-initio protocol combined with NMR restraints, we obtained a final structure where the NRAS$^{Q61K}$ orientation within the HLA-A*01:01 groove agreed completely with the experimentally

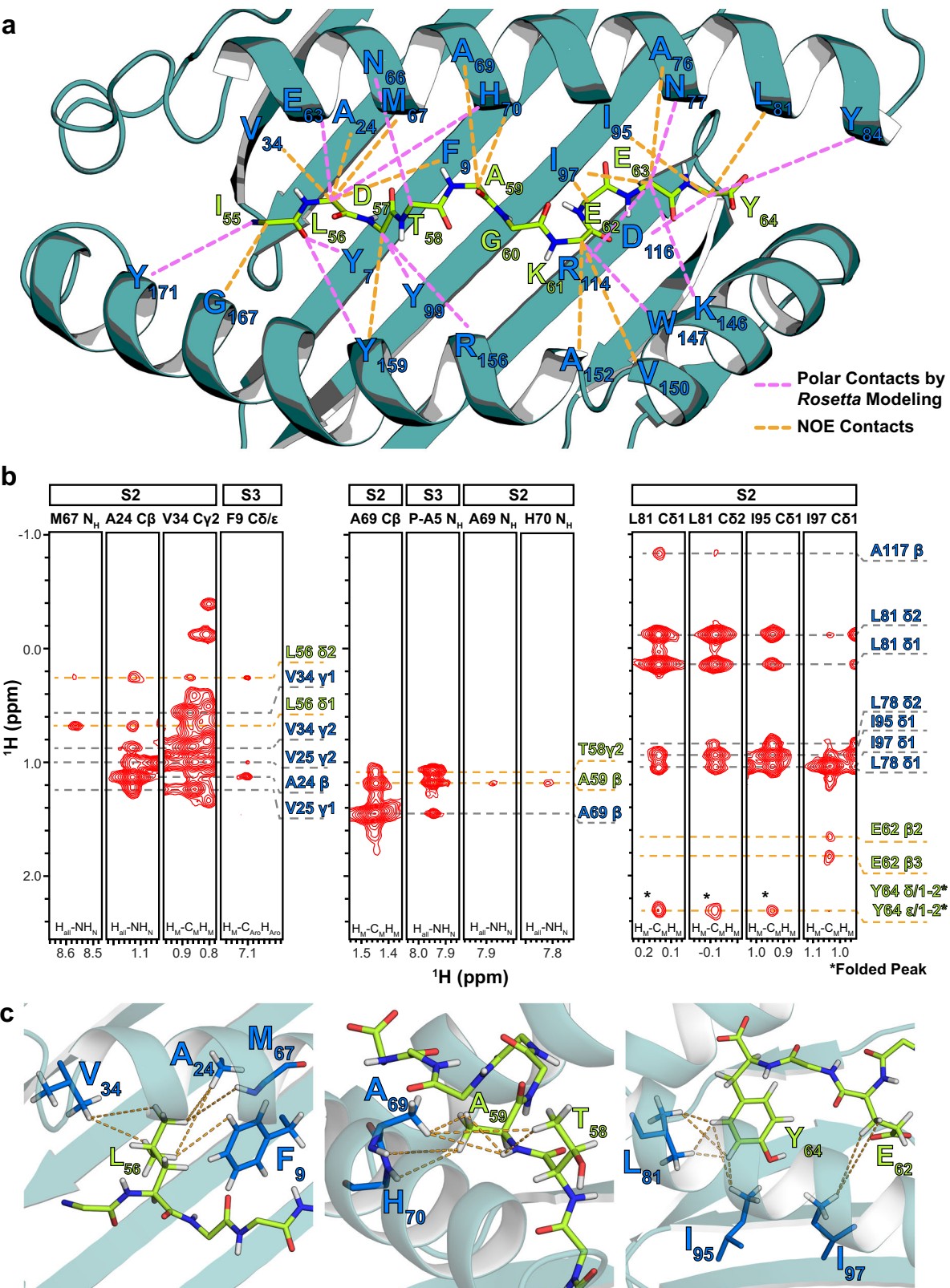

derived NOE constraints (Figs. 2a and 3b). The 10 lowest energy bound NRAS^Q61K conformations provided a well-defined ensemble (backbone RMSD of 0.61 Å and all-atom RMSD of 1.01 Å) (Fig. 3f). Notably, the overall Rosetta energy of the resulting decoys was much lower when NMR restraints were used (Fig. 3c, e), highlighting that experimental restraints were essential for guiding peptide docking towards the native structure. Together, these results support that a sparse set of

NMR distance restraints, obtained using selective methyl/aromatic-labeled samples, are both necessary and sufficient for accurate modeling of the NRAS^Q61K/HLA-A*01:01/hβ₂m complex solution structure.

To characterize neoantigen features that are relevant for interactions with receptors, we examined the orientation of the NRAS^Q61K peptide relative to the HLA-A*01:01 groove and accessible molecular surface[12]. We focused on the representative lowest-energy structure, as

**Fig. 2 | Intermolecular NOE contacts observed between the HLA-A*01:01 groove and NRAS^Q61K.** **a** Intermolecular NOEs observed between HLA-A*01:01 residues and NRAS^Q61K residues (dotted orange lines) in 3D SOFAST NOESY experiments are shown with HLA-A*01:01 groove (dark green) and peptide residues (green) labeled. Intermolecular polar contacts not observed experimentally, but observed in the final structure following high-resolution refinement using the Rosetta FlexPepDock ab-initio protocol are also shown (dotted purple lines). **b** Representative $^1H_M$-$^1H_M$ NOESY strips corresponding to intra- (gray dotted lines) and inter-molecular (orange dotted lines) NOE contacts, shown for regions spanning the HLA-A*01:01 groove (A/B pocket – left, B/C pocket – center, and E/F pocket – right). The top of

the strip is labeled with the name of the sample used (S2 or S3) and the residue/atom name corresponding to the strip. The bottom of the strip is labeled with the NMR experiment where the NOE strip was obtained. Atoms corresponding to HLA-A*01:01 are labeled blue, while atoms corresponding to NRAS^Q61K are labeled green. NOE cross-peaks that are folded due to sweep width restrictions are indicated with an asterisk (*). **c** View of intermolecular NOE contacts (dotted lines) observed in panel B with HLA-A*01:01 (dark green) and peptide (light green) side chains shown as sticks. Our final refined structure of the complex (PDB ID 6MPP) was used to generate structure views.

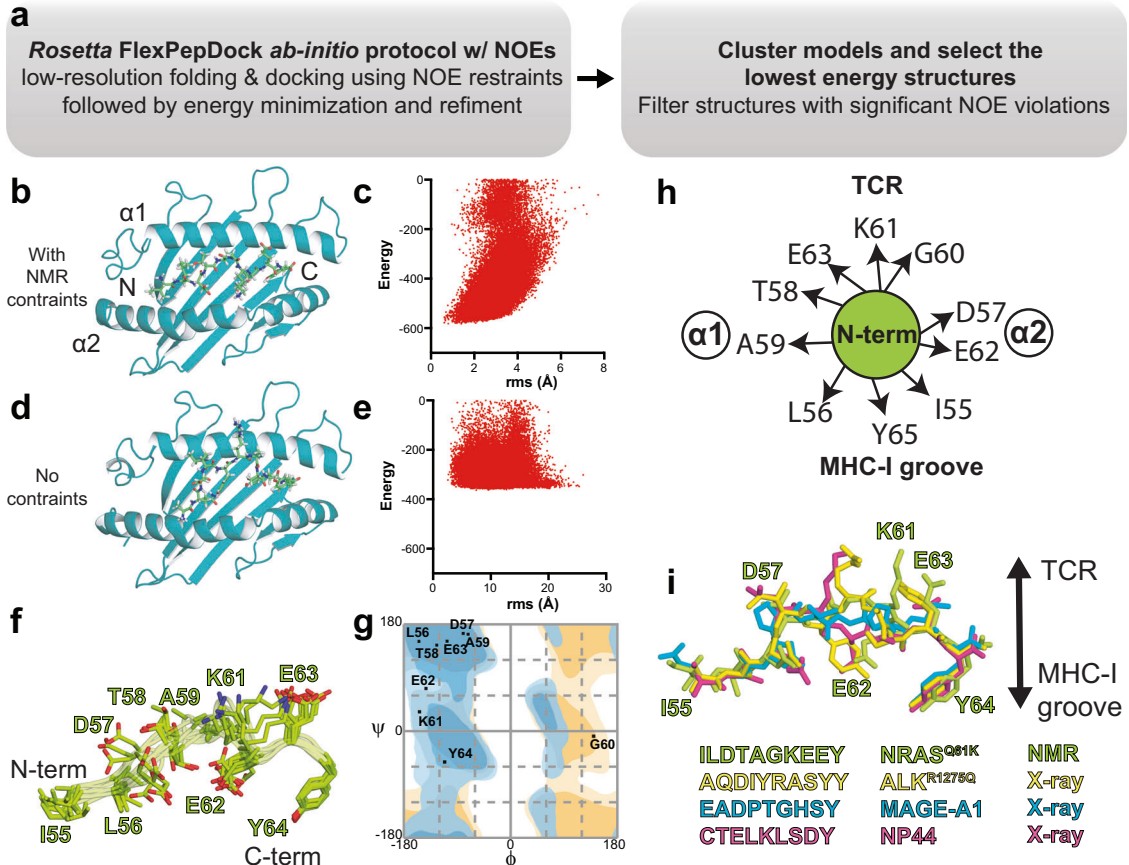

**Fig. 3 | Molecular surface features of the NRAS^Q61K/HLA-A*01:01 groove for TCR recognition.** **a** Flow chart of the combined NMR and Rosetta FlexPepDock ab-initio protocol. **b**, **d** The Rosetta FlexPepDock ab-initio protocol was performed **b** with and **d** without NMR restraints. The HLA-A*01:01 groove is shown in dark green and NRAS^Q61K is shown in light green. The peptide cannot properly dock into the HLA-A*01:01 groove without NMR restraints due to geometric restrictions imposed by the MHC-I pockets. **c**, **e** Energy landscape plots showing energy (Rosetta energy units) versus backbone heavy atom root mean squared deviation (r.m.s., Å) to the lowest energy structure are shown for 50,000 models of FlexPepDock ab-initio protocol performed with **c** NOE-derived NMR constraints (Supplementary Table 2) and **e** without NMR constraints. **f** Overlay of the ten lowest energy NRAS^Q61K

conformations in complex with HLA-A*01:01/$β_2$m (not shown for clarity). **g** Ramachandran plot showing the φ/ψ angles of each residue of NRAS^Q61K peptide bound to HLA-A*01:01. The φ/ψ were computed on the peptide in the lowest energy structure HLA-A*01:01 bound NRAS^Q61K structure using the RAMPAGE server. Blue – general amino acid favored/allowed. Orange – glycine favored/allowed. **h** Cartoon schematic of the side chain orientation of the NRAS^Q61K (bound to HLA-A*01:01) displayed to TCRs as viewed from the N-terminal end relative to the MHC-I α1/α2 helices. **i** Comparison of the conformation and surface features of NRAS^Q61K (PDB ID 6MPP) displayed by HLA-A*01:01 to TCRs with peptides taken from X-ray structures of other known HLA-A*01:01-restricted epitopic peptides (ALK^R1275Q - PDB ID 6AT9, MAGE-A1 - PDB ID 3BO8, NP44 - PDB ID 4NQV).

there were no major differences among the NMR ensemble of HLA-A*01:01 bound NRAS^Q61K conformations (Fig. 3f). A Ramachandran plot of the HLA-A*01:01 bound NRAS^Q61K revealed backbone dihedral angles within the expected favored/allowed regions (Fig. 3g). NRAS^Q61K was oriented such that the side chains of G60, K61, and E63 formed an exposed molecular surface, whereas those for I55, L56 and Y64 were buried in the MHC-I pockets (Fig. 3h, i). Since the K61 side chain is solvent-exposed, the divergent electrostatic surface features of NRAS^Q61K relative to the wild-type NRAS^Q61 peptide can be exploited for targeted recognition using autologous or engineered

immunoreceptors. Finally, we performed a superposition of the NRAS^Q61K peptide conformation with X-ray structures of other known immunodominant HLA-A*01:01 epitopes. The peptide backbone conformation of the bound NRAS^Q61K was compared with other peptide/HLA-A*01:01 complexes, such as ALK^R1275Q (AQDIYRASYY – PDB ID 6AT9), MAGE-A1 (EADPTGHSY – PDB ID 3BO8), and NP44 (CTELKLSDY – PDB ID 4NQV). While the conformation of the peptide N- and C-termini was conserved and anchored through interactions within the A/B- and F-pockets of HLA-A*01:01, significant divergence was observed between residues 4 and 7 (Fig. 3i), revealing a unique surface

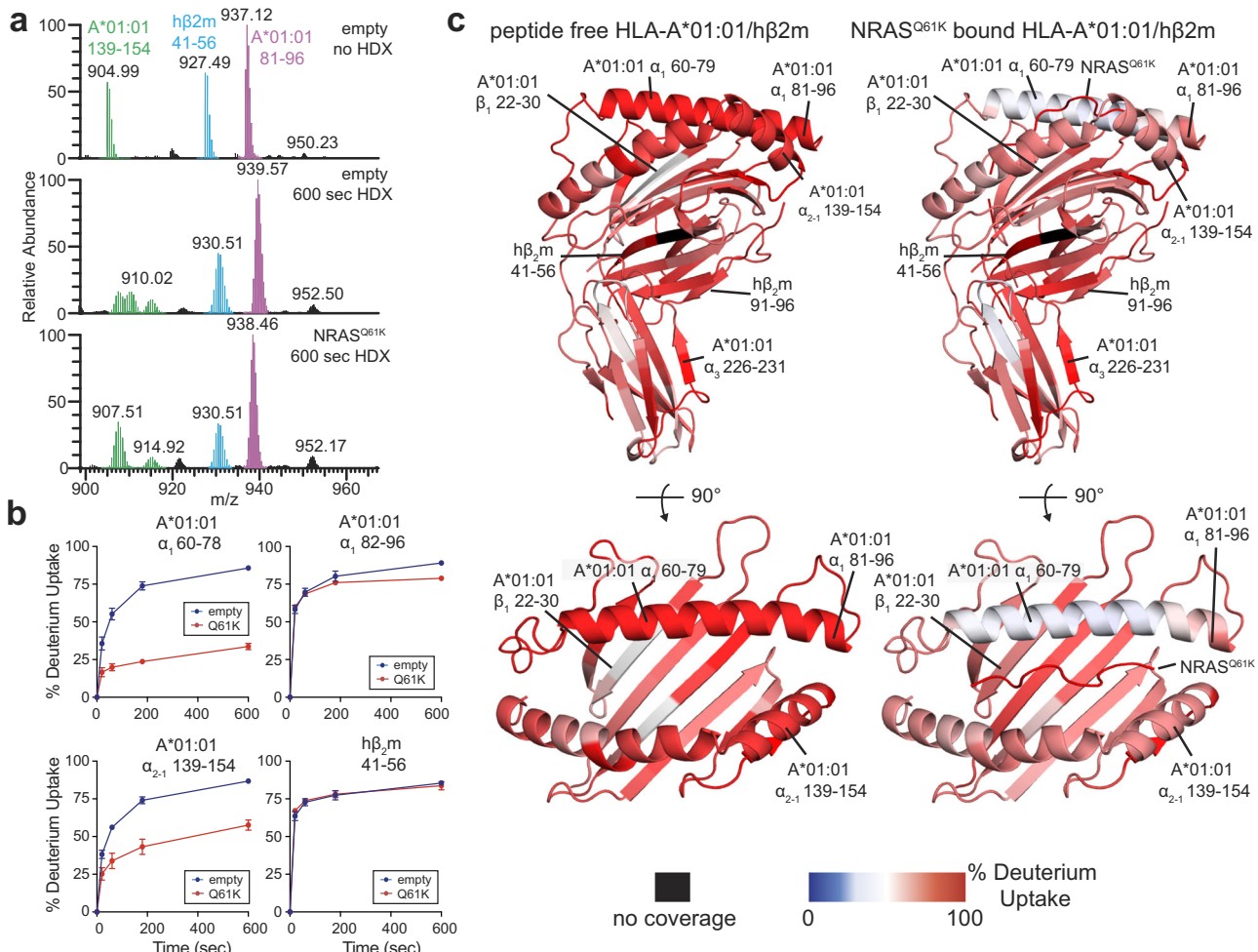

**Fig. 4 | Global conformational plasticity of empty vs NRAS$^{Q61K}$ bound HLA-A*01:01/hβ$_2$m visualized by hydrogen/deuterium exchange. a** Mass spectrometric envelopes for peptide fragments comprising residues HLA-A*01:01 81–96 (magenta), HLA-A*01:01 139–154 (green), and hβ2m 41–56 (blue) for emptied HLA-A*01:01 with no HDX (*i.e.*, all H sample) and 600 s HDX compared with NRAS$^{Q61K}$ bound HLA-A*01:01 at 600 s HDX. The peptide fragment masses are noted. **b** Kinetic graphs of % deuterium uptake (back-exchange corrected) for different peptide fragments as a function of HDX time (0, 20, 60, 180, or 600 s) shown for emptied (blue) versus NRAS$^{Q61K}$ bound (red) HLA-A*01:01/hβ$_2$m. Data are mean % deuterium uptake ± SD from triplicate experiments. **c** Structure view of average % deuterium uptake at 600 s (back-exchange corrected) for empty and NRAS$^{Q61K}$ bound HLA-A*01:01/hβ$_2$m plotted onto PDB IDB 6MPP without or with atoms corresponding to NRAS$^{Q61K}$. Color ranges from deep blue (no deuterium uptake) to red (100% deuterium uptake). Black indicates regions where peptides were not obtained. Mean % deuterium uptakes at 600 s (back-exchange corrected) were resolved to individual peptide fragments and obtained from three independent samples.

chemistry and geometry for NRAS$^{Q61K}$ presented to T-cell repertoires, relative to other known HLA-A*01:01 presented epitopes.

## NRAS$_{55-64}$ epitopes show differences in solvent accessibility and conformational plasticity

To compare the conformation and solvent accessibility of the MHC-I groove in empty versus NRAS$^{Q61K}$-bound HLA-A*01:01/hβ$_2$m, we performed hydrogen/deuterium exchange (HDX) followed by proteolysis and mass spectrometry analysis of the resulting peptide fragments[68,69]. Compared to control experiments performed in the absence of deuterium, HDX performed on empty and NRAS$^{Q61K}$-bound HLA-A*01:01/hβ$_2$m samples for 600 s resulted in different levels of deuterium uptake for peptide fragments spanning the entire HLA-A*01:01 and hβ$_2$m molecules (Fig. 4a–c, Supplementary Table 5, Supplementary Data 1–3). Analysis of percent deuterium uptake as a function of HDX reaction time revealed saturation of most peptide fragments within the 600 s timescale and distinct profiles between empty and NRAS$^{Q61K}$ bound HLA-A*01:01/hβ$_2$m (Fig. 4b, Supplementary Data 2 and 3). Specifically, regions corresponding to α$_1$ helix (residues 60–78 and 82–96) and α$_{2-1}$ helix (residues 139–155) showed significant differences

in HDX behavior between empty versus NRAS$^{Q61K}$ bound states (Fig. 4b, c, Supplementary Figs. 8 and 9, Supplementary Data 3). In contrast, the HLA-A*01:01 α$_3$ domain and hβ$_2$m showed similar HDX rates in the absence or presence of peptide (Fig. 4b, c, Supplementary Fig. 9). Together, our HDX results demonstrate that the absence of NRAS$^{Q61K}$ peptide destabilizes the HLA-A*01:01 groove, consistently with our DSF and NMR experiments (Fig. 1b, d).

The Q61K mutation represents only a subset of cancer-associated RAS$_{55-64}$ epitopes. Thus, we sought to compare the assembly of HLA-A*01:01 with alternative NRAS$_{55-64}$ neoepitopes commonly found in tumors, such as Q61R, Q61L, and Q61H, relative to wild-type Q61 and Q61K[20,70]. Incubation of TAMRA-dye labeled NRAS$^{Q61K}$ with unlabeled NRAS$^{Q61K}$/HLA-A*01:01/hβ$_2$m complex and varying concentrations of unlabeled competitor (wild-type or residue 61 mutant NRAS$_{55-64}$ peptide) revealed IC$_{50}$ values between 225 and 329 nM, demonstrating mutation at residue 61 does not drastically alter the binding affinity of NRAS$_{55-64}$ epitope with HLA-A*01:01 (Fig. 5a–c, Supplementary Fig. 10a–e), as supported by our NMR structure where the side chain for residue 61 does not form any interactions with groove residues (Fig. 3f–i). In agreement, DSF experiments revealed that complexes

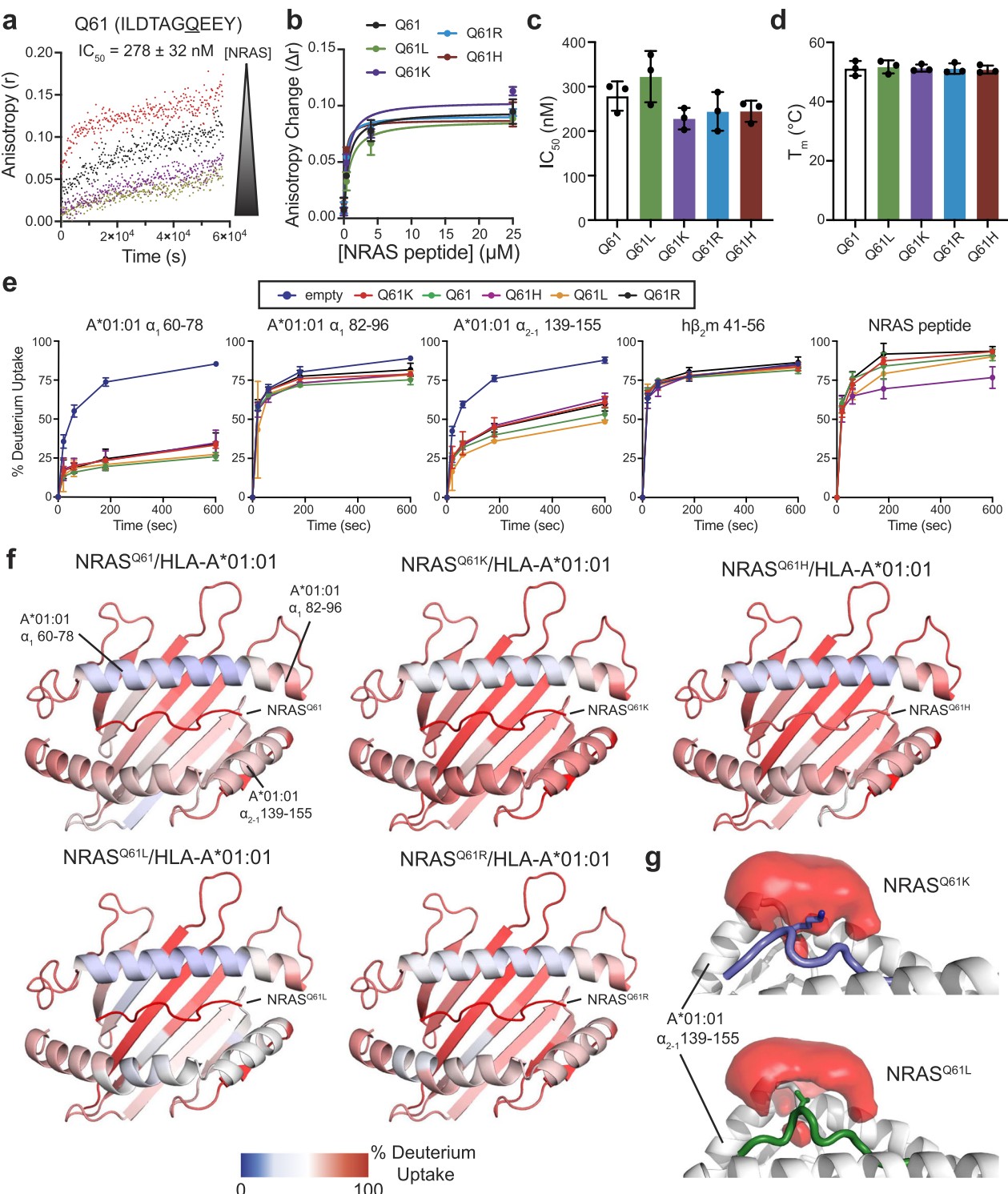

**f** NRAS^Q61^/HLA-A*01:01   NRAS^Q61K^/HLA-A*01:01   NRAS^Q61H^/HLA-A*01:01

NRAS^Q61L^/HLA-A*01:01   NRAS^Q61R^/HLA-A*01:01

formed between HLA-A*01:01/hβ₂m and wild-type or mutant NRAS₅₅₋₆₄ peptides exhibited similar thermal stabilities in the 51 to 53 °C range (Fig. 5d, Supplementary Fig. 10f). Together, these data provide experimental evidence that HLA-A*01:01 can present a panel of naturally occurring Q61 neoepitopes of RAS₅₅₋₆₄ with similar binding affinities.

We next examined whether Q61 mutations influence the conformational plasticity of the peptide/HLA-A*01:01/hβ₂m complexes using our established HDX assay (Fig. 5, Supplementary Figs. 11 and 12). Regions corresponding to the HLA-A*01:01 C-terminal end of the α₁ helix (residues 82–96) and hβ₂m (residues 41–56) exhibited similar HDX rates for HLA-A*01:01 complexes prepared with different

NRAS₅₅₋₆₄ neoepitopes (Fig. 5e, Supplementary Fig. 8). In contrast, we observed differences in deuterium uptake for HLA-A*01:01 bound to different NRAS₅₅₋₆₄ neoepitopes for specific regions of the HLA groove, including the α₁ helix (residues 60–78) and the α₂₋₁ helix (residues 139–155). In particular, the α₂₋₁ helix of NRAS^Q61L^/HLA-A*01:01 exhibited significantly reduced deuterium uptake (more protection) compared to NRAS^Q61H^ and NRAS^Q61R^ bound samples, while NRAS^Q61K^/HLA-A*01:01 exhibited highest deuterium uptake among the peptide-loaded samples (Fig. 5e, f, Supplementary Fig. 13). Peptide fragments corresponding to the NRAS peptide showed similar deuterium uptake levels of approx. 90%, with the exception of NRAS^Q61H^, which showed

**Fig. 5 | Affinity, stability, and conformational plasticity of cancer specific NRAS₅₅₋₆₄ neoepitopes within the HLA-A*01:01 groove. a** Fluorescence anisotropy (r) of 25 nM TAMRA-NRAS^Q61K in the presence of 4 µM unlabeled NRAS^Q61K/HLA-A*01:01/hβ₂m and varying concentrations of the specific unlabeled competitor NRAS Q61 peptide. [NRAS] shown in µM units are 0.0 (red), 0.25 (black), 4.0 (purple), 25 (yellow). The data were analyzed by global fitting using Dynafit 4 (http://www.biokin.com/dynafit). Estimated IC₅₀ value for NRAS Q61 binding to HLA-A*01:01/hβ₂m is noted. Data are mean ± SD for *n* = 3 independent experiments. **b** Comparison of anisotropy change (Δr) for different NRAS₅₅₋₆₄ Q61 mutants as a function of competitor concentration extracted from competition fluorescence anisotropy experiments. Data are mean ± SD for *n* = 3 independent experiments. **c** Summary of fluorescence anisotropy measured IC₅₀ values for different NRAS₅₅₋₆₄ Q61 mutants competing for binding to NRAS^Q61K/HLA-A*01:01/hβ₂m. Data are mean ± SD for *n* = 3 independent experiments. **d** Summary of DSF measured thermal stability values (T_m, °C) for different NRAS₅₅₋₆₄ Q61 mutants in complex with

HLA-A*01:01/hβ₂m. Data are mean ± SD for *n* = 3 technical replicates. **e** Kinetic graphs of % deuterium uptake (back-exchange corrected) for different peptide fragments as a function of HDX time (0, 20, 60, 180, or 600 s) shown for emptied (blue) versus HLA-A*01:01/hβ₂m bound to different NRAS Q61 mutant peptides. Data are mean % deuterium uptake ± SD from biological triplicates. **f** Structure view of average % deuterium uptakes for peptide fragments of HLA-A*01:01/hβ₂m bound to different NRAS Q61 mutant peptides at 600 s (back-exchange corrected) plotted onto PDB IDB 6MPP. Color ranges from deep blue (no deuterium uptake) to red (100% deuterium uptake). Mean % deuterium uptakes at 600 s (back-exchange corrected) were resolved to individual peptide fragments and obtained from biological triplicates. **g** Average water occupancy maps across 3 × 1 µs MD simulations for NRAS^Q61K/HLA-A*01:01 and NRAS^Q61L/HLA-A*01:01 complexes generated using VolMap tool in VMD and visualized in PyMOL v2.5.2 with isosurface contour level 0.2.

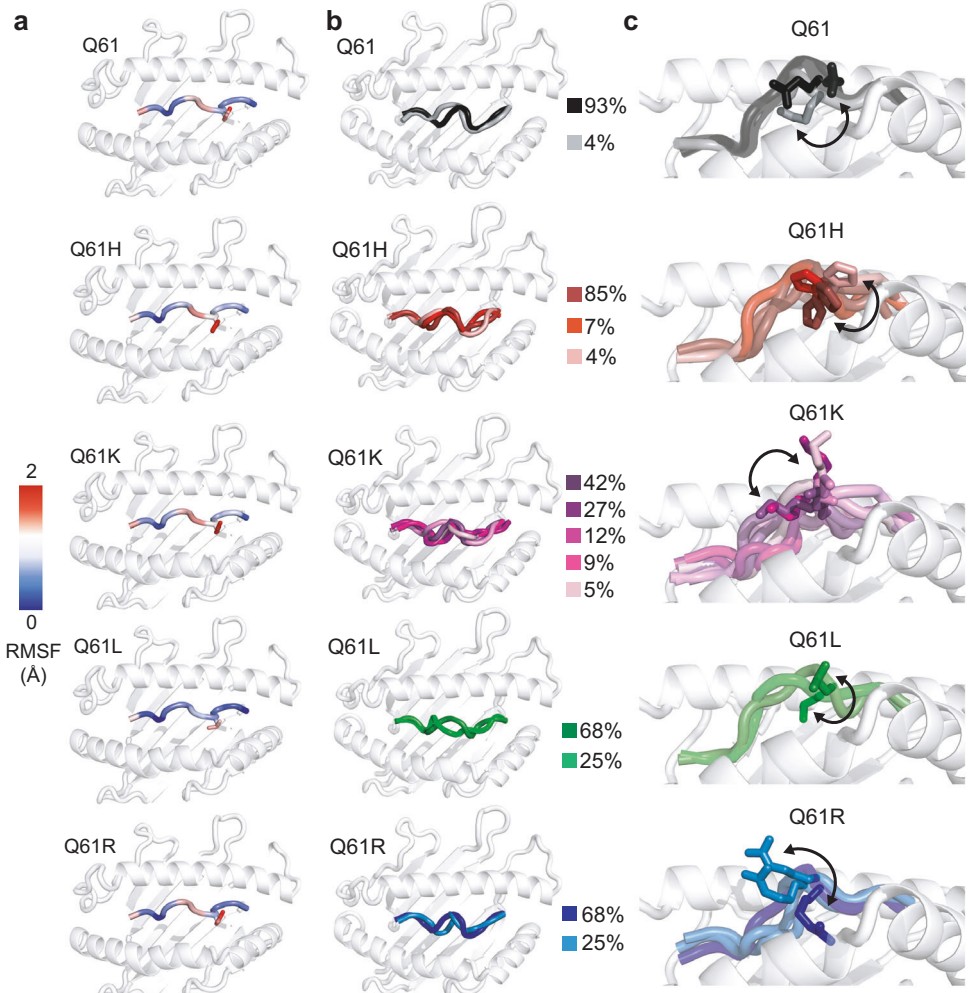

**Fig. 6 | MD simulations of NRAS₅₅₋₆₄ neoepitope/HLA-A*01:01 complexes highlight differences in peptide backbone and side chain conformational landscapes. a** Root mean square fluctuation (RMSF, Å) of C_α atoms (for the peptide backbone, shown as colored cartoon) and heavy atoms (for residue 61, shown as sticks) averaged across 3 × 1 µs MD simulations. **b** Peptide backbone clusters generated using a backbone root mean squared deviation (RMSD, Å) cut-off of 0.5 Å. The percentage of frames corresponding to each cluster relative to the total number of frames across the three replicate trajectories (3000) is noted. **c** Peptide side chain clusters generated from the backbone clusters in (**b**) using a heavy atom RMSD cut-off of 1 Å. Arrows denote movement of the side chain for residue 61. In **a**–**c** the HLA-A*01:01 groove is shown in light gray cartoon for reference.

increased protection with % deuterium uptake levels of approx. 75% (Fig. 5e, f, Supplementary Fig. 8). To obtain a mechanistic basis for our observed differences in deuterium uptake for NRASQ61K versus Q61L in HDX experiments, we performed 3 independent 1 µs all-atom molecular dynamics (MD) simulations in explicit solvent for HLA-A*01:01 bound to NRASQ61K or Q61L peptides, and compared solvent

occupancies near the HLA-A*01:01 α₂₋₁ helix. Visualization of water occupancy maps revealed that, in contrast to Q61K, the hydrophobic side chain of Q61L was stably packed against the α₂₋₁ helix, with reduced exposure to solvent molecules (Fig. 5g), in agreement with increased HDX protection levels of the α₂₋₁ helix (Fig. 5f). Thus, our HDX data and MD-derived water occupancy maps reveal different

solvent accessibility profiles for different NRAS Q61 variants, which underscore changes in both the average conformation and dynamics of captured NRAS$_{55-64}$ neoepitopes within the HLA-A*01:01 groove.

Finally, to examine how Q61 mutations influence the conformational landscape of NRAS$_{55-64}$ peptide within the HLA-A*01:01 groove, which dictates recognition by TCRs or designed antibodies[7,71], we complemented our MD simulations of HLA-A*01:01 bound to NRAS$^{Q61K}$ or NRAS$^{Q61L}$ with $3 \times 1\,\mu s$ MD trajectories of HLA-A*01:01 bound to either NRAS$^{Q61}$, NRAS$^{Q61H}$, or NRAS$^{Q61R}$ (9 μs in total), all modeled by threading to our NMR structure of NRAS$^{Q61K}$/HLA-A*01:01. The major conformation of NRAS$^{Q61K}$ obtained in MD simulations exhibited deviations from the NMR/docking-derived model, as would be expected for a dynamic peptide/protein complex (Supplementary Table 4). We computed average all heavy atom root mean square fluctuation (RMSF) values, which revealed differences in local backbone and side chain fluctuations for each peptide (Fig. 6a). In particular, the peptide backbone and side chain of Q61L were significantly more conformationally restricted than other mutants, while Q61K displayed the greatest mobility. We then characterized specific backbone and side chain motions contributing to the global conformational ensemble of each peptide. First, we computed order parameters (S$^2$) for backbone φ (phi) and ψ (psi) dihedral angles, which quantify the degree of conformational restriction relative to an internal coordinate system. S$^2$ values range from 0 (conformationally unrestricted) to 1 (rigid) (Supplementary Fig. 14). These analyses demonstrated that the backbone of Q61L is significantly more rigid relative to the other peptides, with a single trajectory showing increased mobility for the backbone of G60. Likewise, χ$_1$ side chain dihedral angles determined across MD trajectories showed variations in side chain dynamics between different NRAS$_{55-64}$ peptides (Supplementary Fig. 15). We performed single-linkage clustering, which allowed us to extract representative backbone conformations for each peptide (Fig. 6b). For each backbone cluster, we also performed a second round of clustering using all heavy atoms, generating backbone-specific side chain clusters (Fig. 6c). In agreement, we observed a reduction of conformational space sampled by the Q61L peptide, which can be described by only two major clusters of side chain conformations relative to a larger number of clusters for the other peptides (3, 3, 4, and 5 for Q61, Q61R, Q61H, and Q61K, respectively) (Fig. 6c). Taken together, these results reveal distinct conformational ensembles sampled by NRAS$_{55-64}$ neoepitopes, which would differentially prime each peptide for specific interactions with TCRs. Specifically, our HDX experiments and MD simulations show that the Q61L neoepitope side chain is less solvent-exposed due to stable packing against the α$_{2-1}$ helix, whereas the Q61K and Q61R side chains are oriented toward the TCR interface. While this could limit accessibility to TCRs, we speculate that the observed conformational rigidity of the Q61L peptide in the pMHC state could also reduce the entropy loss upon pMHC-I/TCR complex formation, leading to a more favorable binding free energy.

### NRAS$^{Q61K}$ binding across the HLA allelic landscape displays distinct features to TCRs

The HLA haplotypes of different patients influence their eligibility for pHLA-targeted therapies because only a subset of HLA allotypes can display neoepitope targets of interest[16]. To expand the set of known alleles that can display NRAS$^{Q61K}$, we first predicted binding against a total of 10,386 HLA-A-, B-, C-, E-, and G- alleles with netMHCpan-4.1[41]. The top predicted binders for each classical class Ia were HLA-A*01:191, HLA-B*05:237, and HLA-C*05:188, which exhibited distinct antigen groove chemistries relative to HLA-A*01:01, based on a sequence alignment of groove residues (Fig. 7a, x-axis). Despite divergent groove amino acid compositions, the computationally derived peptide binding motifs of HLA-A*01:191, HLA-B*05:237, and HLA-C*05:188 resembled HLA-A*01:01, where Asp and Tyr are favored at residues 3 and 10 of the peptide, respectively (Fig. 7b). To examine how groove

chemistry contributes to peptide stabilization from a structural perspective, we generated RosettaMHC models[26,34] of NRAS$^{Q61K}$ bound to different HLA alleles using our NRAS$^{Q61K}$/HLA-A*01:01 structure as a template, and compared structural and energetic features derived from each model. This approach allowed us to evaluate whether the NRAS$^{Q61K}$ peptide can adopt a similar overall backbone conformation to that seen in our NRAS$^{Q61K}$/HLA-A*01:01 structure. NRAS$^{Q61K}$/HLA-A*01:191 models exhibited identical peptide/HLA interface energies, low clash scores, and formed a similar network of peptide/HLA contacts relative to the NRAS$^{Q61K}$/HLA-A*01:01 structure (Fig. 7c, d). In contrast, NRAS$^{Q61K}$/HLA-B*05:237 and NRAS$^{Q61K}$/HLA-C*05:188 models showed higher peptide/HLA interface energies and larger clash scores due to steric clashes introduced by amino acid polymorphisms in the MHC-I groove (Fig. 7c, d). Together, our in silico modeling suggests that NRAS$^{Q61K}$ forms a similar structure in complexes formed with HLA-A*01:01 and HLA-A*01:191, but likely adopts a different conformation when bound to HLA-B*05:237 and HLA-C*05:18. In vitro refolding of HLA-A*01:191 with β$_2$m in the presence of NRAS$^{Q61K}$ peptide generates a good yield of NRAS$^{Q61K}$/HLA-A*01:191/β$_2$m protein complexes, with an enhanced thermal stability, T$_m$, of 58.5 °C, relative to the HLA-A*01:01 complex (Supplementary Fig. 16). The experimental validation further confirmed our computational prediction that NRAS$^{Q61K}$ is also a high-affinity peptide binder to HLA-A*01:191. Therefore, any peptide-focusing immunoreceptor of NRAS$^{Q61K}$ in the context of HLA-A*01:01 should exhibit some level of cross-reactivity with NRAS$^{Q61K}$/HLA-A*01:191, while re-engineering of receptors is likely to be required for targeting the same neoepitope displayed by either HLA-B*05:237, or HLA-C*05:188.

To further identify and characterize HLA allotypes that can be targeted using NRAS$^{Q61}$-focused therapeutics, we sorted the 1,248 predicted NRAS$^{Q61K}$ binders according to their observed global HLA frequency obtained from the HLA Allele Frequency Net Database[72]. HLA-A*01:01 was the only strong binder of NRAS$^{Q61K}$ that is represented in >10% across different ethnic groups, with a global HLA frequency of approx. 13.2% (Fig. 7e). However, several allotypes represented across all populations, such as HLA-C*04:01 (12.3%), HLA-B*15:01 (5.7%), HLA-C*08:02 (2.8%), were predicted as weak binders by netMHCpan (Fig. 7f). Analysis of peptide/HLA interface energies and clash scores of structural models[34,35] revealed similar trends in the predicted conformation of NRAS$^{Q61K}$ across different HLA complexes (Fig. 7e, f). To validate our in silico predictions, we performed in vitro refolding of NRAS$^{Q61K}$ with HLA-B*15:01 and HLA-C*08:02, which exhibited decamer peptide binding motifs that are consistent with the NRAS$^{Q61K}$ sequence (Fig. 7g). In agreement, NRAS$^{Q61K}$ assembled with both HLA-B*15:01 and HLA-C*08:02, as shown in SEC traces, albeit at lower efficiencies relative to HLA-A*01:01 (Fig. 7h, left). Purified HLA-B*15:01 and HLA-C*08:02 proteins with NRAS$^{Q61K}$ exhibited moderate thermal stabilities in the 51 to 52 °C range, similar to the HLA-A*01:01 complex (Fig. 7h, right). Together, these results highlight the potential of NRAS$^{Q61K}$ as a public neoepitope target for cancer immunotherapy. Our analysis further reveals sets of common HLA allotypes, which can present NRAS$^{Q61K}$ in a similar overall conformation, suggesting that they can be targeted using a single therapeutic modality to maximize patient cohort coverage.

## Discussion

Identification of RAS neoepitopes displayed on the surface of tumors and recognized by T cells or engineered therapeutics in an HLA-restricted manner has opened up new avenues for targeted cancer immunotherapy[21,27,70,73–79]. Efficient engineering of chimeric antigen receptors (CARs), TCRs, and directed antibodies with enhanced affinity and specificity requires detailed characterization of neoepitope/MHC-I complexes, together with receptor sequences (i.e. from TCR repertoire analysis[80]) and evaluation of cytotoxic tumor killing[18,23]. While landmark studies defining TCR repertoires and peptide/HLA

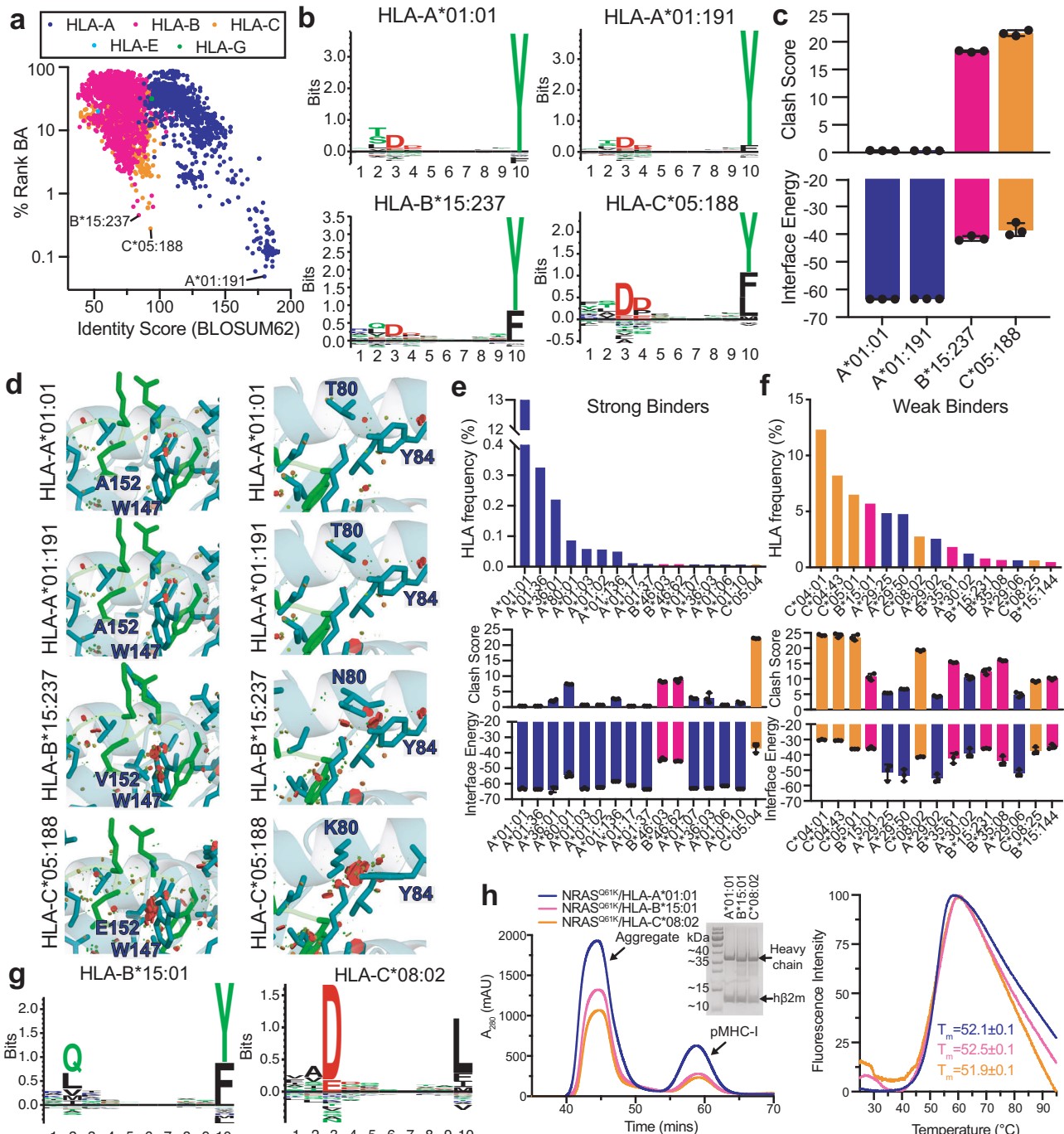

**Fig. 7 | In silico and in vitro evaluation of the HLA binding repertoire of NRAS^{Q61K}. a** In silico prediction of NRAS^{Q61K} (ILDTAGKEEY) binding to 10,386 different HLA molecules with netMHCpan-4.1. The % rank binding affinity (BA) from netMHCpan-4.1 is compared to the BLOSUM62 identity score for all HLA alleles relative to HLA-A*01:01, calculated for HLA groove residues. The top predicted NRAS^{Q61K} binders for each HLA-A-, B-, and C- class are noted. **b** Decamer peptide binding motifs (generated by Seq2Logo) for the top predicted NRAS^{Q61K} binders for each HLA-A-, B-, and C- class. **c** Interface energy (REU) obtained from RosettaMHC models of NRAS^{Q61K} with the top predicted binders for each HLA-A-, B-, and C-classes and their MolProbity clash score. Our solved NMR structure (PDB ID 6MPP) was used as the template for modeling. Data are mean ± SD for $n = 3$ independent experiments. **d** Zoom view of RosettaMHC models of NRAS^{Q61K} with the top predicted binders for each HLA-A-, B-, and C- class. HLA residues within 3.5 Å of the peptide are shown. Van der Waals overlaps or steric clashes are represented with

red/green discs. **e** and **f** Top: HLA frequency (%, obtained from the Allele Frequency Net Database for all populations) for the top predicted **e** strong and **f** weak NRAS^{Q61K} binders. Bottom: Interface energy (REU) obtained RosettaMHC models of NRAS^{Q61K} with the most frequently represented strong and weak binders in the global population and their MolProbity clash scores. Data are mean ± SD for $n = 3$ independent experiments.
**g** Decamer peptide binding motifs (generated by Seq2Logo) for HLA-B*15:01 and HLA-C*08:02, which are predicted as weak binders of NRAS^{Q61K}. **h** Left: SEC traces of in vitro refoldings of NRAS^{Q61K} with HLA-A*01:01/hβ2m, HLA-B*15:01/hβ2m, and HLA-C*08:02/hβ2m. The SDS-PAGE gel of the purified pMHC-I complexes is also shown. Right: DSF experiments performed on the purified pMHC-I complexes with fitted thermal stability (T_m, °C) shown. Data are mean ± SD for $n = 3$ technical replicates.

mediated tumor killing have highlighted the potential of targeting NRAS[Q61K] as a robust avenue for cancer therapy[25,27,29], studies on the assembly, conformational plasticity, and structure of NRAS[Q61K]/HLA complexes are currently lacking. Here, we provide mechanistic insights into how the decameric NRAS[Q61K] peptide (ILDTAGKEEY), as well as related RAS[55-64] derived neoepitopes, assemble with and stabilize the HLA-A*01:01/h$\beta_2$m complex. Furthermore, using NRAS[Q61K]/HLA-A*01:01 as a model of a moderate affinity peptide that is refractory to X-ray crystallography, we outline an integrative pipeline for detailed characterization of the solution structure, conformational plasticity and HLA binding repertoire of clinically relevant neoepitope/HLA targets with desired features for cancer immunotherapy.

Our DSF, NMR, and HDX data support that NRAS[Q61K] and h$\beta_2$m act in concert to stabilize and shape the conformation of the HLA-A*01:01 groove (Figs. 1a–f, 4b, c), in agreement with previous reports for analogous pMHC-I systems[60,61]. We find that association between NRAS[Q61K] and empty HLA-A*01:01 requires several hours to reach saturation according to our NMR spectra, suggesting that HLA molecules undergo a slow transition to a peptide-receptive conformation (Fig. 1b, c, g–i). Alternative NRAS Q61 neoepitopes serving as clinical targets, such as NRAS Q61L, Q61R, and Q61H[21,70], also assembled with HLA-A*01:01 with similar affinity, kinetics, and thermal stabilities, relative to NRAS[Q61K] (Fig. 5a–d). Notably, HDX and MD simulations provide evidence that the precise amino acid at position 61 impacts the conformational ensembles sampled by both the NRAS[55-64] peptide and the HLA-A*01:01 groove (Figs. 5e–g, 6). Our results further reveal that NRAS[Q61K] can associate with common HLA alleles other than HLA-A*01:01, such as HLA-B*15:01 and HLA-C*08:02 (Fig. 7), thereby expanding the cohort of patients for which NRAS[Q61K]-targeted immunotherapies can be applied. Like NRAS[Q61K]/HLA-A*01:01, NRAS[Q61K]/HLA-B*15:01 and NRAS[Q61K]/HLA-C*08:02 complexes failed to form crystals for X-ray diffraction studies. Thus, future structure determination of these complexes using our integrative NMR-based approach provides an attractive avenue to validate our computational modeling results. One limitation of this study is that pMHC-I molecules were prepared lacking membrane association and glycosylation that are known to have important implications for peptide loading in vivo. Specifically, MHC-I glycosylation at the conserved Asn86 position influences interactions with molecular chaperones and other quality control molecules of the APP pathway, which mediate peptide repertoire optimization (Tapasin, TAPBPR, UGGT1), and could therefore influence cancer antigen selection and presentation[81–84].

Our integrated approach highlights the powerful combination of sparse NMR data and integrative computational modeling using Rosetta[65,85–87]. This methodology will be particularly useful for cases in which pMHC-I and pMHC-II complexes fail to crystallize, or where in silico structural modeling[35] is prohibited due to lacking of a structural template[35], such as dynamic pMHC-I complexes, or non-canonical complexes displaying longer peptides up to 15 amino acids[32,88,89]. Our method also has the potential to be adapted and applied for structural characterization of a range of other peptide/proteins systems, for example PDZ-peptide and SH3-peptide interactions that play a role in cellular signaling, and trafficking[90]. To evaluate the performance of state-of-the-art structure prediction methods on a similar system, we compared our structure to an AlphaFold-Multimer[91] derived model of the NRAS[Q61K]/HLA-A*01:01 complex (Supplementary Figs. 17 and 18). We observed a C$_\alpha$ RMSD of 2.82 Å between the experimental and predicted NRAS[Q61K] structures. While AlphaFold-Multimer correctly placed the NRAS[Q61K] P2 and P10 anchoring residues within the HLA-A*01:01 groove, the predicted model deviated significantly from our solved structure at residues 4–7, which includes the mutation site at position 61 (Supplementary Fig. 17). The incorrect placement of these key side chains in the predicted model hinders their use to guide therapeutics development, or vaccine design efforts, and underscores a need for experimental validation of computationally derived models,

as also shown in our recent study of a pMHC-targeting chimeric antigen receptor[92].

Previous studies have focused on HLAs presenting clinically relevant RAS[5-14] neoepitopes, which contain driver mutations at positions 12 and 13 that are most commonly represented in HRAS and KRAS[21,70]. For example, the structures of KRAS[5-14] G12V bound to HLA-A*02:01[22] and KRAS[5-14] G12D bound to HLA-C*08:02[93] have provided valuable information towards development of engineered T cells and antibodies targeting pMHC-I surface markers. Our study addresses the RAS[55-64] epitopic peptide, which encompasses Q61 mutants most commonly represented in HRAS and NRAS[21,70]. The overall structure of the NRAS[Q61K]/HLA-A*01:01/h$\beta_2$m complex reveals a different surface chemistry for NRAS[Q61K] where G60, K61 and E63 side chains are primed to engage cognate TCRs (Fig. 3h, i). HDX experiments and MD simulations highlight differences in the conformational landscape of residue 61 for Q61, Q61K, and Q61R (Figs. 5 and 6). These findings could provide a basis for modeling previously identified TCRs, such as N17.3.2 and N17.5, shown to specifically recognize NRAS[Q61K]/HLA-A*01:01 complexes[27]. Our results also point to a strategy for designing neoantigen-selective receptors that avoid interactions with conformational ensembles found in the wild-type protein, while optimizing favorable interactions with the neoepitope pMHC. Specifically, we find that Q61K and Q61R have basic side chains oriented toward the TCR/scFv interface, which are conformationally distinct from wild-type Q61 as well as mutants Q61H and Q6L. These results have further implications for recognition of the pMHC-I surfaces by TCRs and therapeutic antibodies[7,71]. The information gleaned from our structure and MD simulations can be leveraged for modeling TCR/pMHC-I and antibody/pMHC-I complexes, or to engineer synthetic receptors with enhanced affinity and specificity. Taken together, our results provide a view of pMHC-I complexes as dynamic ensembles, providing a toehold for understanding antigen immunogenicity and cross-reactivity with TCRs.

## Methods
### Protein expression and purification
DNA encoding the luminal domain of the class I MHC-I heavy chain HLA-A*01:01, HLA-B*15:01, HLA-C*08:02, and the light chain human $\beta_2$m (h$\beta_2$m) was graciously provided by the NIH Tetramer Core Facility. For the HLA-A*01:01 plasmid, a stop codon was introduced by PCR before the BirA tag due to known interference with NMR and X-ray crystallography experiments. BirAStop Forward Primer: 5′-ctg tct tcc cag ccc tga tcc ctg cat cat att - 3′; BirAStop Reverse Primer: 5′-aat atg atg cag gga tca ggg ctg gga aga cag - 3′. DNA sequences corresponding to NRAS[Q61K] peptide (ILDTAGKEEY) were prepared as fusion constructs with an N-terminal a His$_6$ tag, immunoglobulin-binding domain of streptococcal protein G (GB1) solubility tag, and enterokinase cleavage site (DDDDK) subcloned in pET22b vector[49]. Plasmids were transformed into Escherichia coli BL21(DE3) (Novagen) and then expressed in LB-broth (natural isotopic abundance samples) or 1× M9 minimal media (for isotopically labeled samples) at 37 °C. MHC-I heavy and light chain molecules were expressed separately as inclusion bodies, solubilized in 6 M guanidine-HCl/0.1 mM dithiothreitol (DTT), mixed together with antigenic peptide for refolding and then purified. NRAS[Q61K] peptide was prepared from the His$_6$-GB1-NRAS[Q61K] fusion construct by purification with Ni-NTA Agarose affinity chromatography followed by cleavage of NRASQ[61K] from His$_6$-GB1 with enterokinase followed by a final round of purification using a Aeris PEPTIDE XB-C18 column (3.6 µm, 150 × 4.6 mm, Phenomenex), with cleavage and purity validated by mass spectrometry. For in vitro refolding, 10 mg of peptide and 200 mg mixtures of heavy chain: light chain at a 1:3 molar ratio were slowly diluted over 24 h into refolding buffer (0.4 M Arginine-HCl, 2 mM EDTA, 4.9 mM reduced L-glutathione, 0.57 mM oxidized L-glutathione, 100 mM Tris pH 8.0) at 4 °C while stirring. Refolding proceeded for four days at 4 °C without stirring.

Purification of pMHC-I complexes was performed by size exclusion chromatography (SEC) with a HiLoad 16/600 Superdex 75 pg column at 1 mL/min with running buffer (150 mM NaCl, 25 mM Tris pH 8). For HLA-A*01:01 complexes used in crystallization trails a second round of purification occurred via anion exchange chromatography with a mono Q 5/50 GL column at 1 mL/min using a 40 min 0–100% gradient of buffer A (50 mM NaCl, 25 mM Tris pH 8) and buffer B (1 M NaCl, 25 mM Tris pH 8). Proteins were extensively dialyzed into 150 mM NaCl, 25 mM Tris pH 8 prior to crystallography screens.

### Generation of empty HLA-A*01:01/hβ2m complexes

HLA-A*01:01/hβ2m complexes were emptied of bound NRAS$^{Q61K}$ peptide by brief exposure to pH 12.5 followed by SEC as previously described[54]. Briefly, 60 µL of 400 µM NRAS$^{Q61K}$/HLA-A*01:01/hβ2m (purified following in vitro refolding) was incubated for 30 min at 4 °C. 10 µL of 0.2 M sodium phosphate pH 12.5 was then added to the sample at 4 °C and incubated for 10 min. Samples were then spun at 13,000 r.p.m. for 10 min at 4 °C. Samples were immediately loaded onto a Superdex 200 Increase 10/300 GL column at 0.5 mL/min with running buffer (50 mM NaCl, 20 mM sodium phosphate pH 7.2) for purification at 25 °C.

### Liquid chromatography–mass spectrometry (LC–MS)

Validation of emptied and NRAS$^{Q61K}$ bound HLA-A*01:01 samples (Supplementary Fig. 4) was achieved by LC–MS. LC–MS was carried out with passage through a Higgins PROTO300 C4 column (5 µm, 100 mm × 2.1 mm) followed by electron ion spray (ESI) mass spectroscopy performed on a Thermo Finnigan LC–MS/MS (LTQ). Analysis and deconvolution of LC–MS data were performed using MagTran.

### Sample preparation for NMR

For all NMR samples, protein expression was achieved by induction with 1 mM isopropyl-D-thiogalactoside (IPTG) at an $OD_{600}$ of 0.6 followed by cell growth at 37 °C for 4 h at 200 r.p.m. Purified isotopically labeled HLA-A*01:01 inclusion bodies from 1× M9 preparations in D$_2$O were refolded with NRAS$^{Q61K}$ and hβ2m with isotopic labeling schemes as noted below. Following refolding and purification pMHC-I samples were exhaustively buffer exchanged into NMR buffer (50 mM NaCl, 20 mM sodium phosphate pH 7.2, 0.01% (v/v) NaN$_3$ and 1U Roche protease inhibitor cocktail in 90% H$_2$O/10% D$_2$O). Isotope labeling schemes for samples 01 through sample 04 are provided in Supplementary Table 1 and below:

Sample S1 In sample S1, HLA-A*01:01 is Ile $^{13}$CH$_3$ for δ$_1$ only; Leu $^{13}$CH$_3$/$^{12}$C$^2$H$_3$; Val $^{13}$CH$_3$/$^{12}$C$^2$H$_3$ labeled in an otherwise $^2$H, $^{13}$C, $^{15}$N background (herein referred to as ILV* labeled) where both NRAS$^{Q61K}$ and hβ2m are at natural isotopic abundance. Val and Leu side chains exhibit linearized $^{13}$C spin systems that allow efficient magnetization transfer for larger proteins in solution[94]. Sample S1 is used for 3D experiments that permit both backbone assignment (3D HNCA, 3D HN(CA)CB, 3D HNCO) and methyl assignment (3D HMCM[CG]CBCA)[94]. The U-[$^{15}$N, $^{13}$C, $^2$H] (Ile $^{13}$CH$_3$ for δ1 only; Leu $^{13}$CH$_3$/$^{12}$C$^2$H$_3$; Val $^{13}$CH$_3$/$^{12}$C$^2$H$_3$) ILV* labeled sample was prepared in 1× M9 minimal media in D$_2$O supplemented with 1 g/L $^{15}$NH$_4$Cl and 3 g/L $^{13}$C, $^2$H glucose (Sigma #552151). Methyl labeling was achieved for Ile and Leu/Val, respectively, with addition of 70 mg/L 2-ketobutyric acid-$^{13}$C$_4$,3,3-$^2$H$_2$ (Sigma #607541) and 120 mg/L 2-keto-3-(methyl-d$_3$)-butyric acid-1,2,3,4-$^{13}$C$_4$, 3-$^2$H (Sigma #637858) added 1 h prior to induction. NRAS$^{Q61K}$ and hβ2m were natural abundance. Sample S1 is also referred to as ILV* labeled HLA-A*01:01.

Sample S2 In sample S2, Ala, Ile, Leu, and Val methyl groups of HLA-A*01:01 are labeled in an otherwise $^2$H, $^{12}$C, $^{15}$N background (herein referred to as AILV labeled) where both NRAS$^{Q61K}$ and hβ2m are at natural isotopic abundance. Sample S2 enables measurement and assignment of NOE cross-peaks with high sensitivity and resolution through the collection of methyl-methyl, amide-methyl, and amide-amide 3D SOFAST NOESY experiments, in addition to conventional filtered/edited NOESY methods[63,64]. The U-[$^2$H, $^{12}$C, $^{15}$N] Ala $^{13}$Cβ; Ile $^{13}$Cδ1; Leu $^{13}$Cδ1/$^{13}$Cδ2; Val $^{13}$Cγ1/$^{13}$Cγ2 labeled sample was prepared in 1× M9 minimal media in D$_2$O supplemented with 1 g/L $^{15}$NH$_4$Cl, 3 g/L $^{12}$C, $^2$H glucose (Sigma #552003) and 0.2 g/L $^2$H,$^{15}$N ISOGRO (Sigma #608300). To achieve methyl labeling for Ile, Leu/Val, and Ala methyls, respectively, 70 mg/L 2-ketobutyric acid-4-$^{13}$C,3,3-$^2$H$_2$ (Sigma #589276) and 120 mg/L 2-keto-(3-methyl-$^{13}$C)-butyric-4-$^{13}$C,3-$^2$H acid (Sigma #589063) were added 1 h prior to induction, while 100 mg/L L-alanine-3-$^{13}$C,2-$^2$H (Sigma #740055) was added 30 min prior to induction. NRAS$^{Q61K}$ and hβ2m were natural abundance. Sample S2 is also referred to as AILV labeled HLA-A*01:01.

Sample S3 In sample S3, HLA-A*01:01 is labeled with AILV methyls in addition to Phe and Tyr aromatic side chains in an otherwise $^2$H, $^{12}$C, $^{15}$N background (herein referred to as FYAILV labeled)[63] where hβ2m is U-[$^2$H] and NRAS$^{Q61K}$ peptide is U-[$^{15}$N] labeled. Sample S3 allows for unambiguous assignment of intermolecular NOEs between NRAS$^{Q61K}$ and HLA-A*01:01 without interference of signals arising from HLA contacts with hβ2m. The U-[$^2$H, $^{12}$C, $^{15}$N] Ala $^{13}$Cβ; Ile $^{13}$Cδ1; Leu $^{13}$Cδ1/$^{13}$Cδ2; Val $^{13}$Cγ1/$^{13}$Cγ2; Phe $^{13}$C, $^{15}$N; Tyr $^{13}$C, $^{15}$N labeled sample was prepared in 1× M9 minimal media in D$_2$O supplemented with 1 g/L $^{15}$NH$_4$Cl, 3 g/L $^{12}$C, $^2$H glucose (Sigma #552003) and 0.2 g/L $^2$H,$^{15}$N ISOGRO (Sigma #608300). To achieve methyl labeling for Ile, Leu/Val, and Ala methyls, respectively, 70 mg/L 2-ketobutyric acid-4-$^{13}$C,3,3-$^2$H$_2$ (Sigma #589276) and 120 mg/L 2-keto-(3-methyl-$^{13}$C)-butyric-4-$^{13}$C,3-$^2$H acid (Sigma #589063) were added 1 h prior to induction, while 100 mg/L L-alanine-3-$^{13}$C,2-$^2$H (Sigma #740055) was added 30 minutes prior to induction. To achieve aromatic labeling for Phe and Tyr, 50 mg/L U-[$^{13}$C,$^{15}$N]-Phe (Sigma #608017) and 50 mg/L U-[$^{13}$C,$^{15}$N]-Tyr (Sigma #607991) were added 1 h prior to induction. NRAS$^{Q61K}$ was U-[$^{15}$N] while hβ2m was U-[$^2$H, $^{14}$N, $^{12}$C]. Sample S3 is also referred to as AILVFY labeled HLA-A*01:01 with $^{15}$N NRAS$^{Q61K}$.

### NMR spectroscopy

For NMR chemical shift assignments of HLA-A*01:01, we utilized a multipronged approach previously applied to the β-chain-labeled B4.2.3 TCR[43]. Specifically, we acquired a suite of TROSY-based triple-resonance measurements (3D HNCA, 3D HN(CA)CB, 3D HNCO) and a suite of 3D SOFAST NOESY measurements (Supplementary Table 1) recorded at 800 MHz at 25 °C[63]. 3D HNCA, 3D HN(CA)CB, 3D HNCO experiments had acquisition times of 30 ms ($^{15}$N), 20 ms ($^{13}$CO) and 10/5 ms ($^{13}$Cα /$^{13}$Cβ) in the indirect dimension. Backbone amide assignments were validated by acquiring amide-amide NOES using 3D H$_N$-NH$_N$ and 3D N-NH$_N$ SOFAST NOESY experiments. AILV methyl resonances were assigned using a 3D HMCM[CG]CBCA methyl out-and-back experiment[94] recorded on 0.5 mM pMHC-I samples at 800 MHz at 25 °C. AILV methyl assignments were validated and stereo-specifically disambiguated by acquiring methyl-methyl NOEs with 3D H$_M$-C$_M$H$_M$ and 3D C$_M$-C$_M$H$_M$ NOESY experiments. Backbone amide and AILV methyl assignments were cross-validated by acquiring methyl-to-amide NOEs using 3D H$_N$-C$_M$H$_M$ and C$_M$-NH$_N$ experiments. Phe and Tyr aromatic side chains were assigned using C$_{Aro}$-C$_M$H$_M$, C$_{Aro}$-NH$_N$, H$_M$-C$_{Aro}$H$_{Aro}$, and H$_N$H$_{Aro}$-NH$_N$ 3D NOESY experiments[63]. Free state resonance assignments of NRAS$^{Q61K}$ were obtained from 2D $^1$H-$^1$H NOESY and $^1$H-$^1$H TOCSY spectra of 1 mM NRAS$^{Q61K}$ (natural isotopic abundance). Assignments of the bound state were obtained with H$_N$H$_{Aro}$-C$_M$H$_M$ and H$_N$H$_{Aro}$-NH$_N$ 3D NOESY experiments using Sample S3. Intramolecular (HLA-A*01:01 to HLA-A*01:01) and intermolecular (HLA-A*01:01 to NRAS$^{Q61K}$) nuclear Overhauser effects (NOEs) were using experiments summarized in Supplementary Table 1 and Supplementary Table 2.

For NMR experiments to determine conformational changes in the HLA groove in the empty and peptide-loaded states, 2D $^1$H-$^{13}$C SOFAST HMQC spectra of 50 µM emptied AILV labeled HLA-A*01:01/

natural abundance $h\beta_2m$ in the absence or presence of 50 μM $NRAS^{Q61K}$ peptide were recorded at a $^1H$ field of 800 MHz at 25 °C. In all samples 150 μM of excess natural isotopic abundance $h\beta_2m$ added for stabilization of the complex. Chemical shift deviations (CSP, ppm) were determined using the equation: $\Delta\delta^{CH3} = [1/2(\Delta\delta H^2 + \Delta\delta C^2/4)]^{1/2}$ for AILV methyls. The change in peak intensity was determined by calculating the ratio of $I_{empty}/I_{peptide}$ bound from peak intensities extracted from NMRFAM-SPARKY. V34, L179, and A140 methyl groups were not used for either CSP or intensity analysis due to complete line broadening in the empty state (signal-to-noise <10).

For assembly kinetics experiments, 2D $^1H$-$^{13}C$ SOFAST HMQC experiments were recorded for 100 μM $^{15}N$ labeled $NRAS^{Q61K}$ in the free state. Next, 2D $^1H$-$^{13}C$ SOFAST HMQC spectra were acquired in 1 h time increments following the addition of 300 μM emptied (natural abundance) HLA-A*01:01/$h\beta_2m$ at a $^1H$ field of 800 MHz at 25 °C. At each time point, the changes in amide signal of free $NRAS^{Q61K}$ resonance relative to the sample without HLA-A*01:01 were fit with a single-phase exponential decay in GraphPad Prism to determine the half-life ($t_{1/2}$).

Relaxation delays (d1) for non-SOFAST experiments were set to 1.2 to 1.5 s, and 0.2 s in SOFAST experiments. Data were processed using nmrPipe[95] and analyzed using NMRFAM-SPARKY[96].

### Rosetta FlexPepDock ab-initio protocol

**Create the initial structure of the complex.** Initial starting structure of the unbound HLA-A*01:01/$h\beta_2m$ was obtained from the atomic coordinates of a previously solved HLA-A*01:01/$h\beta_2m$ presenting a decameric peptide with the atomic coordinates corresponding to the peptide removed (PDB ID 6AT9). The initial $NRAS^{Q61K}$ backbone conformation of the peptide was assumed to be extended and was created using the BuildPeptide Rosetta utility.

```
BuildPeptide.default.linuxgccrelease-database
~/rosetta/Rosetta/main/database -in:file:fasta nrasq61k.
fasta -out:file:o nrasq61k_extended.pdb
```

The initial $NRAS^{Q61K}$ peptide was then positioned manually near (-2.5 Å) the HLA-A*01:01 groove with an orientation of N to C of ILD-TAGKEEY based on the known P2 and P8 anchor positions for the HLA-A*01:01 groove using PyMOL. The orientation of the peptide was derived from our NOE measurements. Finally, the PDB file was created in agreement with the specifications in the original FlexPepDock protocol (the peptide coordinates should be the last model)[65].

**Initial files and structures.** The unbound HLA-A*01:01/$h\beta_2m$ and HLA-A*01:01/$h\beta_2m$ + $NRAS^{Q61K}$ structures were "relaxed" and "prepacked", respectively. The lowest energy initial structures were selected for restraint guided docking.

Command line for "Relax":

```
/path/to/rosetta/binaries/relax.linuxgccrelease
-relax:constrain_relax_to_start_coords  -relax:coord_
constrain_sidechains -relax:ramp_constraints false -s
your_structure.pdb
```

Command lines for "Prepack":

```
/path/to/rosetta/binaries/FlexPepDocking.linuxgc
crelease -s "best_start_relaxed.pdb" -ex1 -ex2aro -use_
input_sc -unboundrot "best_native_relaxed.pdb" -flexpep_
prepack -nstruct 1
```

### FlexPepdock protocol.

```
Production runs
qsub -pe orte 300 production.job
```

```
Content of production.job file
#
#
# JOB TEMPLATE FOR BAKER CLUSTER
#
# Job name
#$ -N FlexPepDock_log_ab_init
#
# User current working directory
#$ -cwd
#
# Don't combine input and output
#$ -j y
#
echo ""
echo $PATH
echo ""
echo "Executing on: $HOSTNAME"
echo "Number of hosts operating on: $NHOSTS"
echo "Number of queued slots in use for parallel job:
$NSLOTS"
# where $NSLOTS is as submitted to pe command to the OpenGrid
SGE scheduler
echo ""
echo "Running on $NSLOTS cpus …"
#
#Current Rosetta binaries
EXE = /path/to/rosetta/binaries/FlexPepDocking.mpi.
linuxgccrelease
CMDLINE = "@flags/flags_A01_Nras_ab"
echo $EXE
echo $CMDLINE
mpiexec -n $NSLOTS $EXE $CMDLINE
exit
Flag file "flags_A01_Nras_ab":
#io flags:
-s pdbs/start.pdb
-native pdbs/native.pdb
-out:file:silent_struct_type binary
-out:file:silent decoys.abinit.silent
-scorefile score.flexpepdock.abinit.sc
-out:prefix ab_
#-out:suffix _preopt
#number of structures to produce
-nstruct 50000
#flexpepdock flags:
-lowres_abinitio
-pep_refine
#packing flags
-ex1
-ex2aro
-use_input_sc
-unboundrot pdbs/unbound.pdb
#extra side chain rotamers
-ex1aro
-ex2
-ex3
-ex4
#fragments for peptide
-frag3 frags/frags.3mers.offset
-frag5 frags/frags.5mers.offset
-frag9 frags/frags.9mers.offset
#NOE constraints
-constraints:cst_file constraints/contacts_A01.cst
-constraints:cst_weight 10
```

```
-constraints:cst_fa_file constraints/contacts_A01.cst
-constraints:cst_fa_weight 10
#expert flags
#start the protocol with the peptide in extended conforma
tion (neglect original peptide conformation; extend from
the anchor residue)
-extend_peptide
```

Clustering top rank representatives

```
/path/to/rosetta/binaries/cluster.linuxgccrelease -in:
file:silent decoys.abinit.silent -in:file:silent_struct_
type binary -cluster:radius 2.3 -in::file::fullatom -in:
file:tags tags.of.the.top.500.structures -silent_read_
through_errors -cluster:limit_cluster_size 10 -cluster:
limit_clusters 5
```

## Fluorescence anisotropy

Fluorescence anisotropy ($r$) experiments were performed using a NRAS$^{Q61K}$ peptide labeled with TAMRA-dye (ILDTAGK$^{TAMRA}$EEY, herein called TAMRA-NRAS$^{Q61K}$) (Biopeptek Inc, Malvern, USA). TAMRA association with K61 was modeled to not disrupt interaction with the HLA-A*01:01 groove because K61 is solvent-exposed. For competition experiments, a mixture of purified 4 µM NRAS$^{Q61K}$/HLA-A*01:01/h$\beta_2$m complex and 25 nM TAMRA-NRAS$^{Q61K}$ was treated with graded concentrations of competitor NRAS peptide (either NRAS$^{Q61}$, NRAS$^{Q61K}$, NRAS$^{Q61L}$, NRAS$^{Q61R}$, or NRAS$^{Q61H}$) with concentrations of 0, 0.25, 4 or 25 µM. NRAS peptides were prepared by chemical synthesis (GenScript, Piscataway, USA). Each experiment was performed at 25 °C in a volume of 100 µL and loaded onto a black 96-well polystyrene assay plate (Costar #3915). Data was recorded on a SpectraMax iD5 plate reader with an excitation filter of $\lambda_{ex} = 531$ nm and an emission filter of $\lambda_{em} = 595$ nm with an PMT gain set to high, an integration time of 1200 msec and read height 0.5 mm. Each experiment was performed in triplicate ($n = 3$) and is representative of at least two independent experiments. Experimental values were subtracted from background $r$ values obtained from incubation of TAMRA-NRAS$^{Q61K}$ alone. All samples were prepared in matched buffer (50 mM NaCl, 20 mM sodium phosphate pH 7.2, 0.05% (v/v) Tween-20). IC$_{50}$ values were obtain by fitting to a simple single-site (one phase association) model with DynaFit (http://www.biokin.com/dynafit).

## Differential scanning fluorimetry (DSF)

DSF experiments were performed in triplicate ($n = 3$) on an Applied Biosystems 7900HT Fast Real-Time PCR System with excitation and emission wavelengths set to 470 nm and 569 nm with proteins in buffer of 50 mM NaCl, 20 mM sodium phosphate pH 7.2. Experiments were conducted in MicroAmp Fast 384-well plates with 20 µL total volume containing final concentrations of 7 µM protein and 10 × SYPRO orange dye (Thermo Fisher). Temperature was incrementally increased at a scan rate of 1 °C/min between 25 °C and 95 °C. Data analysis and fitting with a Boltzman sigmoidal curve were performed in GraphPad Prism v9. For experiments with emptied HLA-A*01:01, data were recorded for 7 µM emptied HLA-A*01:01/h$\beta_2$m as well as 7 µM emptied HLA-A*01:01/h$\beta_2$m incubated for different amounts of time with 70 µM NRAS$^{Q61K}$ peptide. The % of empty HLA-A*01:01 was calculated by determining the ratio of the fluorescence intensity of each time point relative to the maximum fluorescence intensity for empty HLA-A*01:01 (no NRAS$^{Q61K}$ peptide added) at 25 °C.

## Molecular dynamics simulations

All-atom molecular dynamics (MD) simulations in explicit solvent were carried out as previously described in GROMACS version 2020.4 using an AMBER99SB-ILDN protein forcefield and TIP3P

water model[49]. LINCS and SETTLE constraint algorithms were used to constrain peptide/protein and water molecules, respectively. An integration time step of 2 fs was used with coordinates output every 10 ps. Short range interactions were treated with a Verlet cut-off scheme with 10 Å electrostatic and van der Walls cutoffs and long-range electrostatics were treated with the PME method with a grid spacing of 1.2 Å and cubic interpolation. Periodic cubic boundaries were used. Simulations were carried out in cubic simulation boxes, and periodic boundary conditions were used in all three spatial dimensions. The thermodynamic ensemble was NPT with temperature kept constant at 300 K by a velocity rescaling thermostat with a stochastic term[97] with 0.1 ps time constant and pressure kept constant at 1 bar pressure using an isotropic Berendsen barostat with 0.5 ps time constant and $4.5 \times 10^{-5}$ bar$^{-1}$ isothermal compressibility. The NRAS$^{Q61K}$/HLA-A*01:01/h$\beta_2$m structure (PDB ID 6MPP) served as the primary input, with the first state used in modeling. For modeling of other NRAS$_{55-64}$ Q61 mutations, Maestro (Schrödinger) was used to generate mutations in the peptide sequence by replacing side chains without any change to the backbone (energy minimization was performed as stated below during the MD simulation). Each pMHC-I system was solvated to overall neutral charge and contained Na$^+$ and Cl$^-$ ions to yield physiological concentration of 0.15 M. Following 500 steps of steepest-descent energy minimization, initial velocities were generated at 65 K with linear heating up to 300 K over 2 ns. Three independent trajectories for each pMHC-I complex were acquired for 1 µs using the Children's Hospital Of Philadelphia Respublica computational cluster. Analysis was performed using GROMACS and the Visual Molecular Dynamics (VMD) package[98]. Visualizations were produced with PyMOL v2.5.2 and gnuplot v5.2.8.

For analysis, periodic boundary conditions were removed and 1000 equally-spaced frames from each trajectory (3000 frames in total for three independent replicates) were extracted in using gmx trjconv. Phi/psi/chi angles and dihedral order parameters were calculated using gmx chi. Peptide per-atom RMSF was calculated for each individual replicate and averaged across replicates using gmx rmsf. Backbone clusters were generated using gmx cluster (single-linkage, RMSD cut-off) on backbone atoms (no oxygen) and an RMSD cut-off of 0.5 Å. Frames corresponding to each cluster were extracted using gmx extract-cluster. Side chain clusters were generated using gmx cluster on protein heavy atoms with an RMSD cut-off of 1.0 Å. Water volume maps were generated in VMD using the VolMap tool, selecting all water atoms within 5 Å of HLA-A*01:01 residues 150-155 and calculating average occupancy with a 1.0 Å resolution and a 1.0 atom size cut-off, and were visualized in PyMOL using isosurface with a contour level of 0.2.

## Hydrogen-deuterium exchange (HDX)

All HDX-MS experiments were designed to follow guidelines recommended by Masson et al.[68] 30 µM stock pMHC-I proteins were dialyzed into HDX Buffer 1 (50 mM NaCl, 20 mM sodium phosphate pH 6.5 in H$_2$O). Emptied MHC-I was prepared using the method outlined in the section "Generation of empty HLA-A*01:01/h$\beta$2m complexes." For reference experiments (i.e., all H or no HDX), 5 µL of 30 µM emptied MHC-I or in vitro refolded pMHC-I was added to 20 µL HDX Buffer 1 and mixed. Next, 25 µL of HDX Quench Buffer (50 mM NaCl, 1 M TCEP (tris(2-carboxyethyl)phosphine), 20 mM sodium phosphate pH 2.35 in H$_2$O) was added and the sample was mixed. The 50 µL samples were immediately injected for LC–MS/MS, which combines pepsin digestion, LC separation using a C8 5 µm column (TARGA, Higgins Analytical, TP-M501-C085), and MS/MS measurement on a Q Exactive Orbitrap Mass Spectrometer (Thermo Fisher Scientific). Following LC–MS/MS experiments for all H samples, peptide fragments corresponding to HLA-A*01:01 (96% sequence coverage), h$\beta$2m (95% sequence coverage), and various NRAS peptides (100% sequence coverage) were identified using Thermo Proteome Discoverer v2.4. For

HDX reactions, 5 µL of 30 µM pMHC-I was added to HDX Buffer 2 (50 mM NaCl, 20 mM sodium phosphate pD 6.5 in $D_2O$), mixed, and incubated for 20, 60, 180, or 600 sec, followed by the addition of 25 µL of HDX Quench Buffer. $D_{frac}$ (fraction D/H) in the final HDX reactions was 0.8. The 50 µL samples were immediately injected into the LC–MS and underwent pepsin digestion, LC separation using a C8 5 µm column, and MS measurement. HDX data points were corrected for back-exchange by performing an additional experiment (*i.e.*, all D) where peptide-free or peptide-loaded samples were heated to 35 °C and 46 °C to induce partial unfolded (5–10% unfolded) for 15 min, respectively, followed by HDX data collection. HDX data points were corrected for back-exchange using Eq. (1)[68]:

$$\%Deuterium\ Uptake = \frac{m_t - m_{0\%}}{m_{100\%} - Dm_{0\%}} \times 100\% \qquad (1)$$

where $m_t$ = peptide centroid mass at a given time point, $m_{0\%}$ is the peptide centroid mass for the non-deuterated peptide obtained from the all H sample, and $m_{100\%}$ is peptide centroid mass for the peptide maximally obtained from the all D sample. HDX reactions were performed at 25 °C followed by pepsin digestion and MS data collection at 0 °C. ExMS2 software was used to identify and analyze deuterated peptides[99]. Percent deuterium uptakes (back-exchange corrected) were resolved to individual peptide fragments and obtained in biological triplicates (three independent in vitro refolding for each protein). Mean % deuterium uptakes for peptide fragments at 600 s were calculated based on triplicate measurements and averaged to each amino acid based on the start and end position and the length of each peptide. The kinetic plots and the B factor for the structure plot were generated by python3 and PyMOL. The Brown-Forsythe and Welch version of one-way ANOVA tests was performed on data corresponding to three biological triplicates using GraphPad Prism to determine statistical significance.

## HLA binding repertoire analysis
netMHCpan-4.1[41] was used to predict binding of NRAS$^{Q61K}$ to a total of 10,386 HLA-A-, B, C, E, and G- alleles. A list of strong and weak binders was obtained together with their % rank binding affinity (BA) from the netMHCpan-4.1 output. BLOSUM62 identity scores[100] relative to HLA-A*01:01 for all 10,386 alleles were generated from a Clustal Omega v1.2.4 alignment of residues representing a pseudo MHC groove (MHC_pseudo.dat file located in $netMHCpan-4.1/ data). Decamer Kullback–Leibler sequence (peptide motif) logos were generated by Seq2Logo[101]. Code used for structural modeling[35] is available on Zenodo[35]. pMHC-I models[26,34] of the complex between NRAS$^{Q61K}$ and all strong or weak binding HLA alleles were generated from the NRAS$^{Q61K}$/HLA-A*01:01 structure template (PDB ID 6MPP) with the following run.sh script, which runs RosettaMHC and then outputs the interface energy (REU) between the peptide and HLA groove: `python3 main.py -nstruct 3 -idealize_relax -relax_after_threading -template_pdb 6MPP.pdb -mhcs mhc_list -peptides pep_list -mhc_chain A -peptide_chain D -pep_start_index 181 -inter- face_cutpoint 180 -out_file nras_HLA_interface_e- nergies.csv` Clash scores were determined with MolProbity. Visualization of clashes in the resulting structures was achieved in PyMOL v2.4.2 (Schrödinger, Inc.) using show_bumps.py (via Thomas Holder - https://pymolwiki.org/index.php/Show_bumps). HLA frequencies were obtained from the HLA Frequency Net Database[72] using the Estimation of Global Allele Frequencies tool for A, B, and C loci with options set to "all" except for Sample Size > 5000.

## AlphaFold-Multimer modeling
An Alphafold-Multimer model for NRAS$_{55-64}$$^{Q61K}$ peptide bound to HLA-A*01:01 was generated using the Google Colab implementation of

Alphafold 2.1.0 in multimer mode (https://colab.research.google.com/ github/deepmind/alphafold/blob/main/notebooks/AlphaFold.ipynb). Amino acid sequences were taken from PDB structure 6MPP, with 3 glycine residues added to each end of the peptide to satisfy the 16 amino acid size constraint.

## Reporting summary
Further information on research design is available in the Nature Portfolio Reporting Summary linked to this article.

## Data availability
The atomic coordinates for the NRAS$^{Q61K}$/HLA-A*01:01/h$\beta$2m structure have been deposited in the Protein Data Bank with PDB ID 6MPP. Other crystal structures used in this study include 6AT9, 3BO8, and 4NQV. NMR chemical shift assignments of HLA-A*01:01 heavy chain have been deposited into the Biological Magnetic Resonance Data Bank (http:// www.bmrb.wisc.edu) under accession number 27632 [doi:10.13018/ BMR27632]. The HDX-MS data have been deposited to the Proteo- meXchange Consortium via the PRIDE[102] partner repository with the dataset identifier PXD044838. The protein sequences and sequence coverage maps for HDX-MS can be accessed from Figshare [10.6084/ m9.figshare.24417373]. The kinetics plots for each peptide fragment of the HDX-MS biological triplicates can be accessed from Figshare for peptide-free [https://doi.org/10.6084/m9.figshare.24415921], NRAS$^{Q61K}$[https://doi.org/10.6084/m9.figshare.24415918], NRAS$^{Q61I}$[https://doi.org/10.6084/m9.figshare.24415942], NRAS$^{Q61H}$ [https://doi.org/10.6084/m9.figshare.24415939], NRAS$^{Q61L}$[https://doi. org/10.6084/m9.figshare.24415945], and NRAS$^{Q61R}$[https://doi.org/10. 6084/m9.figshare.24415948] peptide-loaded HLA-A*01:01/h$\beta_2$m. Source data are provided with this paper.

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

## Acknowledgements

The authors thank Drs. A. Rao, S. Tripathi, and D. Haussler (University of California, Santa Cruz), M. Yarmarkovich, S. Gupta, and J. Maris (Children's Hospital of Philadelphia). This research was supported by NIAID (5R01AI143997), NIDDK (3U01DK112217) and NIGMS (5R35GM125034) grants to N.G.S. The authors acknowledge the Johnson Research Foundation and Dr. L. Mayne (University of Pennsylvania) for assistance with HDX instrumentation. This work was delivered as part of the NextTGen team supported by the Cancer Grand Challenges partnership funded by Cancer Research UK (CGCATF-2021/100002) and the National Cancer Institute (CA278687-01) and the Mark Foundation.

## Author contributions

N.G.S., A.C.M., S.E.G., and D.F-S. designed research. A.C.M., D.F-S., J.S.T., and M.C.Y. prepared recombinant protein samples. A.C.M. and D.F-S. performed NMR experiments, resonance assignments, and data analysis. D.F-S. performed Rosetta flexible docking simulations and solved the NRASQ61K/HLA-A*01:01 complex structure. A.C.M. and J.S.T. recorded DSF, fluorescence anisotropy experiments. A.C.M. performed in silico binding repertoire analysis, molecular modeling, and analyzed data. Y.S. performed HDX-MS experiments and carried out revisions of the manuscript. S.E.G. carried out MD simulations, assisted with HDX data analysis, and prepared the AlphaFold-Multimer model. A.C.M., Y.S., S.E.G., and N.G.S. wrote the paper with feedback from all authors. N.G.S. acquired funding and supervised the project.

## Competing interests

The authors declare no competing interests.
