## [Peer Review File · Nature Communications]

REVIEWER COMMENTS

Reviewer #1 (Remarks to the Author):

Flores-Solis et al. report the use of an integrated structural biology approach to investigate the structure and dynamics of a clinically important neoantigenic peptide, NRAS(Q61K), in HLA-A*01:01. It is an excellent manuscript that I thoroughly enjoyed reading. Each of the major conclusions is robustly supported by multiple lines of evidence, e.g., both MD simulations and HDX experiments support differences in the flexibility of NRAS-Q61 peptide variants in HLA-A*01:01. Although I can't comment on the technical aspects of the NMR or HDX experiments, the biochemical experiments and molecular simulations are technically sound. The manuscript is well-written and easy to follow, and the methods are described in an appropriate level of detail.

The integrated structural biology approach outlined here should be widely applicable, and allow analysis of the structure and dynamics of peptide-HLA complexes not amenable to X-ray crystallography, especially moderate-affinity complexes where the conformation of the peptide may not be well-resolved even if crystals can be obtained. The structures of NRAS(Q61K)/HLA-A*01:01 as well as other pHLA complexes obtained using a similar approach will be useful both for basic immunology and immunotherapy development. I think the paper will also be appreciated by the broader structural biology community as an example of integrated structural modeling. I have only a couple of minor concerns about the agreement between the structures obtained by docking and MD simulations, as discussed below.

Minor comments

1. The authors investigate the assembly of the NRAS(Q61K)/HLA-A*01:01/b2m complex in significant detail and show that binding involves global structural changes in the HLA-A*01:01/b2m complex. However, the relevance of this analysis was not really clear to me. The kinetics and mechanism of peptide binding are presumably very different for the native HLA-A*01:01/b2m complex in vivo and the pH-shocked, refolded HLA in vitro. I suggest to provide some additional discussion of the context and implications of these results.
2. Line 80: "We previously predicted that a public, decameric epitope derived from the recurrent NRAS55-64 mutation, Q61K (ILDTAGKEEY), could interact with the common HLA-A*01:01 molecule." The authors might want to mention explicitly here that the interaction has not only been predicted, but more importantly, experimentally validated in vitro and in vivo.

3. Line 267: "suggesting limited cross-reactivity of TCRs recognizing NRAS(Q61K) with other known epitopes restricted by HLA-A*01:01". This is a major overstatement that should be deleted. Each HLA presents thousands of potentially cross-reactive antigens - comparison of NRAS(Q61K) with three randomly chosen antigens reveals very little about potential cross-reactivity.

4. Line 306: On the significance of differences in deuterium uptake between complexes, could the authors briefly comment (maybe in the methods section) on the criteria for significance? e.g., if only one replicate was done for each complex, how was measurement error estimated.

5. Fig. 6b. This figure shows the representative conformations of each NRAS(55-64)/HLA-A*01:01 variants in MD simulations. However, it is not clear whether the conformations represent genuine flexibility of the peptide, with multiple transitions between each conformation, or alternatively, relaxation of an incorrect initial conformation into a more favorable conformation. Some kind of analysis of RMSD vs time should be shown to distinguish these possibilities.

6. Fig. 6b. It appears that the conformation of NRAS(Q61K)/HLA-A*01:01/b2m modeled by NMR and docking represents only a minor conformation (~5%) in the MD simulations. Does the major conformation observed in MD simulations potentially provide a better model of the NRAS(Q61K)/HLA-A*01:01/b2m complex than that obtained by docking? Does this conformation agree with the experimentally determined NOEs? Since docking does not explicitly account for water, it makes sense that MD could give an improved model.

7. Line 450, fig. S11. The AlphaFold2 model should also be compared with the poses obtained by MD - it may be that the conformation observed in the AlphaFold2 model is also part of the conformational ensemble. Also, is the AlphaFold2 model consistent with the experimental data? Addressing these two points would make the comparison more persuasive.

8. Lines 464-476: This discussion about the importance of conformational flexibility for TCR recognition is slightly overstated, in my opinion. e.g., line 468 - differential recognition of Q61K and Q61 by TCRs could simply be due to the different chemical structure and properties of Gln vs Lys; there's not necessarily a need to invoke dynamics. Line 474, NRAS Q61K and Q61R are proposed to be especially promising neoantigens for antibody targeting due to a distinct conformation compared with the WT peptide - but presumably many neoantigenic peptides show subtle changes in conformation and dynamics compared with the WT peptide - there is no evidence that Q61K and Q61R are unique in this regard.

Minor corrections

1. The reference numbering is off - e.g., ref. 38 at line 128 and ref. 41 at line 143.
2. Line 347: Q61R is stated to sample three conformations, but according to Fig. 6b there are only two conformations. Maybe the legend in Fig. 6b is mislabeled.
3. Line 964 - how was the peptide prepared from the fusion protein?
4. Line 1249 - what model was used for fitting?
5. Line 1268 - what modeling was performed, exactly? Was some kind of energy minimization or docking performed at this stage, or was the side chain simply replaced without any change to the backbone?
6. Fig. S4a/b - y-axis of two top panels should presumably be "relative absorbance". What wavelength was used?
7. Fig. 7d - I suggest moving this panel to a separate figure in the SI - it's difficult to interpret due to its small size.

Reviewer #2 (Remarks to the Author):

Sgourakis and coauthors present an interesting dissection of the binding mode for a neoepitope NRASQ61K to HLA-A*01:01 and the wider HLA allelic landscape, in support of an assessment of this neoepitope's value as an immunotherapy target. This analysis also includes an assessment of how a range of other RAS55-64 neoepitopes bind to HLA-A*01:01. As the authors note, several other studies have defined the TCR repertoires and peptide/HLA-mediated tumor killing associated with NRASQ61K. But they emphasize that studies on the assembly, conformational plasticity, and structure of select NRASQ61K/HLA combinations are lacking. Structural studies have been done for other NRAS neoantigens like RAS5-14, so in this sense the current study completes a key missing piece in the assessment of NRAS for TCR immunotherapy. It does reveal the probable structural uniqueness of NRASQ61K/HLA-A*01:01 and support its use as a target for immunotherapy.

It is a high quality integrative structural modeling study that plugs a hole. The comparative analysis with AF2 Multimer is intriguing, highlighting the ongoing need for structural work in this post-AlphaFold world. The resulting experimentally-generated conformer is sensible and certainly intriguing, but I believe this study belongs in a more specialized journal, given the related body of work that precedes it.

I cannot comment on all the technical elements of the study. I'll restrict detailed comments to the HDX-MS work. Figures 4 and 5 are insufficient to describe the HDX-MS data, and must be expanded.

1. Fig 4c attempts to show a comparative analysis by requiring the reader to compare the “empty” vs NRASQ61K form. Firstly, these should be referred to as the free vs bound samples throughout, and secondly, this is simply too coarse of a comparison to help the reader. Difference plots should be used to support a “delta” deuteration visualization of the datasets.
2. To support this difference plot, the authors must establish the statistical criteria they used to make the assessment of difference. A simple visual of the two kinetics datasets is insufficient. I strongly recommend they follow a recently-published “guidelines” article published in Nature Methods 3 years ago (Masson et al., Nat. Methods 2019, 16, 595-602).
3. These guidelines also contain information on supplementary datasets that should be provided for the reader to gauge the completeness of the study. Note in particular that the mapping of deuteration changes (or kinetics) should only be based on the peptides that survive deuteration analysis, and not simply the sequence map of the undeuterated data.
4. Figure 4a shows a peptide that appears to either demonstrate EX1/EXX kinetics (139-154), overlap, or partial binding. It adds concern over the way data analysis was conducted, as this peptide is indicated to demonstrate a large difference in deuteration upon neoepitope binding (Fig 4b). As the guidelines will indicate, a clearer statement on selection criteria may help, in addition to the provision of all kinetics curves if possible.
5. In connection with this concern, I cannot see what concentration excess of neoepitope to HLA was used in the HDX study. This should be provided as I'm sure it was done to saturate binding (note that upon deuteration there is a high dilution, requiring concentration excess in most cases). And if there was a large excess, how did the authors manage to detect changes in the NRAS peptides themselves? Information is missing.

Reviewer #3 (Remarks to the Author):

Flores-Solis and coworkers combine different biophysical (most notably NMR) and computational methods (Rosetta docking and MD simulations) to determine the solution-state structure of a peptide/human leucocyte antigen (pHLA) complex, NRASQ61K/HLA-A*01:01. This combined integrative computational structural biology approach enabled the authors to determine an atomic-resolution conformational ensemble from a (relatively sparse) set of 25 NOE distance constraints between the peptide and the HLA protein (in addition to a measured dense network of NOE contacts within the HLA protein itself). Using MD simulations, the authors extend their study also to other RAS neoepitopes, such as Q61L, Q61R, and Q61H. Finally, using in-silico and in-vitro approaches, they also examined the HLA binding repertoire of RAS neoepitopes beyond HLA-A*01:01 towards the HLA-B*15:01 and HLA-C*08:02 allotypes.

The amount of work done and the results obtained are impressive. The experiments and computations were carefully done and interpreted, and the main conclusions are supported by the data. Concerning the computations, established state of the art methods were used. However, we have a few requests for further improvements and mutual validation of MD and NMR.

1) The available NMR distance constraints provide a great opportunity for the authors to more deeply scrutinize and carefully validate their MD simulations against their NMR data, by quantifying the NOE restraint violations in their MD simulations of NRAS-Q61K. This analysis is straightforward (for example with gmx disre, in which also proper $1/r^6$ distance averaging for NOE distances is implemented, but it can of course also readily be done differently) and will provide insights into the mutual consistency of MD and NMR. In addition to the overall average violations (global picture), it will be insightful to see which of the 25 NOE-derived distance constraints are violated (if any), by how much (assuming a reasonable upper distance bound of, e.g., 6 Å), and maybe even obtain insights into the reasons underlying the deviations.

2) Which of the 25 NOE contacts are essential for determining the conformation ensemble of the complex, and which ones are not so important? For example, would a few constraints at the N- and C-terminal ends of the peptide (pockets A/B and E/F, respectively) suffice to obtain a similar, well-defined structural ensemble? Using only subsets of the NMR distance constraints for the modeling also enables the authors to use the violations of the unused NOE contacts for cross-validation.

MINOR ISSUES:

1) p. 3, line 133: It should be Ref 37 here, not Ref 40.

2) p. 7, line 321: We would not necessarily agree that the solvent accessibilities are unique. For example, different hydrophobic side chains at position 61 might have very similar accessibilities. What the authors mean there is probably that the solvent accessibilities of the different Q61 variants are DIFFERENT (but not necessarily "unique" in a strict sense).

3) p. 8, line 338: The authors stress the agreement between the S2 order parameters of backbone phi/psi dihedrals and the RMS fluctuations of the backbone atoms. However, this "agreement" is trivial because the backbone RMSF and phi/psi order parameters contain highly redundant information. Backbone atomic fluctuations are largely due to phi/psi dihedral fluctuations, as these are the two most flexible degrees of freedom of the peptide backbone. So the agreement is obvious and should not be overly stressed.

4) p. 8, line 347: The authors discuss the "reduction of conformational space sampled by the Q61L peptide, which can be described by only two clusters relative to the other peptides (3, 3, 4, and 5 for Q61, Q61R, Q61H, and Q61K, respectively) (Fig. 6b,c)." We think this is unclear from Fig. 6b,c because for Q61 and Q61R, for example, there are also only 2 backbone clusters -- but we assume the authors here refer to the larger number of side chain clusters than for Q61L? Could this be clarified in the text, to avoid misunderstandings?

5) p. 8, line 353ff: Are there data to support the reduced entropy loss and/or more favourable binding free energy? If not, these aspects should be clearly marked as speculation.

6) p. 37, line 1263: The sentence "Periodic cubic boundaries were used" sounds odd. We guess what the authors mean here is that the simulations were carried out in cubic simulation boxes, and that periodic boundary conditions were used in all three spatial dimensions?

7) p. 37, line 1264: Can the authors specify what they exactly mean with "V-rescale modified thermostat"? We guess they refer to the velocity rescaling thermostat with a stochastic term by Bussi et al, which is applied via the "v-rescale" mdp option in Gromacs? If yes, please rewrite accordingly and add a reference to Bussi et al.

Reviewer #4 (Remarks to the Author):

In this manuscript by Flores-Solis et al. the properties of MHC class I (MHCI) molecules in complex with a neoantigen from Ras (the NRAS peptide NRASQ61K) is investigated by biophysical and structural means. This question is of importance, since it has become clear that the conformational dynamics either of pMHCI or of the TCR are of critical importance for understanding TCR specificity and affinity. For example, a recent paper in Nature Communications was good evidence for this hypothesis. While the crystal structure of a lower affine TCR and its high affine engineered counterpart showed very little difference in the mode of binding and could not explain affinity enhancement, thermodynamic analysis and MD simulations revealed electrostatic and entropic contributions to the binding free energy (Poole et al., Nat Commun 5333, 2022). In this manuscript, the characterization of the complex is vigorous, employing several biophysical techniques. Empty MHCI is analyzed as well as the neoantigen bound complex. In particular NMR spectroscopy is used to delineate conformational dynamics. Line broadening is severe for the empty MHCI, and as expected spectra become much more disperse and amenable to analysis once peptide is bound. NOE analysis reveals the key interactions in the binding groove and allows to interpret chemical shift changes in the structural context. Moreover, HD exchange in combination with mass spectrometry is utilized to derive the stability of individual secondary structure elements. It becomes evident, that the empty MHCI is largely destabilized. However, two of the

underlying b-strands are stabilized compared to the peptide-bound MHC, indicating that residual stable secondary structure is maintained. Furthermore, different mutants of the NRAS peptide (Q61K, Q61R, Q61H, and Q61L) were probed by the HDX method and complemented by MD simulations. One mutant, Q61L showed decreased solvent accessibility (HDX-derived) and more restricted motions (MD). In contrast the Q61R and A61K mutants show the charged side-chains to be fully exposed and thus represent critical hot spots for the design of engineered TCR's or antibodies.

This is an important manuscript in the area of neoantigen presentation by MHC class I molecules. However, there are several issues that should be addressed prior to publication:

1.) According to the HSQC in Figure 1d, roughly 50% of the MHC is receptive, while the other 50 % need to equilibrate prior to binding the peptide. In contrast in Fig. 1g-i, where the peptide resonances are recorded, no such biphasic behavior is observed. How do the authors explain this discrepancy ? Also, can they say more about the two conformations, e.g. based on their H/D experiments ? Are oligomers of the empty MHCI formed ?

2.) Structural analysis has been performed for the peptide-bound MHCI, but I was wondering whether NOESY spectra were also recorded for the empty MHC molecule and whether this would indicate in more detail which regions are destabilized or stabilized. The H/D exchange data shown later indicate that while most of the binding groove is destabilized, residues 22-30 for example are stabilized. NOE's should be observable for those residues in contact with the neighboring strands. This point relates to the question also asked in 1.) and which concerns the difference between the non-receptive and the receptive MHCI. What are the stabilizing interactions that make the energy barriers towards the receptive conformation so slow ?

3.) H/D exchange is combined with mass spectrometry, yielding very good sequence coverage. What is missing, however, is a Table that clearly denotes the (i) relative deuterium uptake at (ii) what time point and (iii) for which sequence.

4.) When discussing the different HLA alleles, HLA-A*01:191 is pointed out as a high affinity binder, potentially cross-reactive with HLA-A*01:01. It would be important that the authors verify their predictions by experimental evidence, express the A*01:191 allele and measure stability/exchange and to then directly compare, e.g. thermal stabilities. Combining this with for example peptide NMR measurements would indicate whether the observed kinetics of peptide binding is similar for two allotypes, thus allowing to conclude whether the slow conversion of a peptide-free to a peptide-receptive MHC is a conserved phenomenon.

5.) From the further analysis of B*15:01 and C*08:02 alleles it is found that they show very similar stabilities (and thus affinities) in complex with the NRAS mutagenic peptide as the A*01:01 allotype. However, predicted binding affinities are lower. How can this be explained and would it not raise the question how reliable the predictions are?

6.) In perspective, the authors make the claim that antibodies targeting Q61K or Q61R would be a good choice for tumor targeting. However, this is somewhat self-evident given the site of mutation. More interesting is the remark that the dynamic features and conformational differences of the peptides could be exploited by engineering TCR's. Would the authors have a concrete suggestion of how this engineering would profit from their data ? A crystal structure of the TCR bound to the pMHCI might also not solve the issue (see paper by Poole et al., Nat. Commun. 2022), so the dynamic and entropic implications of binding might be critical. A matched pMHCI-TCR tri-molecular modeling approach might therefore be the way to go, so I was wondering about the authors point-of-view regarding this aspect.

Reviewer #5 (Remarks to the Author):

In this manuscript, Floris-Solis et al. describe the structural and dynamic properties of NRAS variants bound to HLA-A0101. The RAS epitopes are of significant interest as they are immunotherapy targets, with potential utilization in cancer vaccine or T cell therapy approaches. The manuscript takes an interesting approach in using nuclear magnetic resonance (NMR) as opposed to crystallography to characterize the Q61K NRAS neoantigen/A0101, followed by hydrogen deuterium exchange, modeling, and simulation to look at how other Q61 variants might lead to altered structures and dynamics.

Major strengths of the manuscript are the high-quality work that is involved and the significant attention to detail. The authors really should be commended for their comprehensive efforts and beautiful presentation of the data. The downside of the paper, however, is that the manuscript is somewhat of a grab-bag of protein biophysics/structural immunology that *generally* confirms much of what is already known, as detailed/demonstrated by NRAS peptides, without new advances for structural immunology or RAS-connected immuno-oncology/immunotherapy.

For example, the manuscript opens with a demonstration that peptide loading follows a slow conformational transition from empty HLA to peptide bound. This has been well known for quite some time. Yes, there is more detail here, and it is nice to see it done with an NRAS peptide, but not much new is learned. Later, we go to findings that different NRAS variants result in different hydrogen-deuterium exchange and (via modeling) different dynamic envelopes. Again, it is already known that small changes to peptides can lead to different HLA and peptide motions and that these can be impactful for immune recognition. While the details to the NRAS peptides are new, there is no reason to

suspect this system would be an exception. Because there is no connection to T cell (or TcR) recognition, peptide potency, vaccination, immunotherapy etc., there is little that is actionable. Rather, what we have is a beautiful biophysical study that confirms what is generally well known. Of course, the use of NMR and H-D exchange, structural modeling, etc. is relatively new to the field, but outside of its impressive technical aspects, I found the paper to be of low potential impact.

Some specific comments:

1) I missed the structural validation of the other variants at position 61. With NMR determination of the K variant, how were the others determined? I think it is all in silico replacement and modeling? This would seem to place a question mark over the interpretations.

2) The argument that peptide dissociation/exchange prevented crystallization seemed handwaving to me. Certainly lower stability complexes have been crystallized. This seemed like an argument to justify the use of NMR in this case.

3) Is it really necessary to show protein production, etc. as supplementary information? I found that additional detail distracting, along with the segue into the protein conformational change. The overall story could use some editing to get to a main point.

REVIEWER COMMENTS

Reviewer #1 (Remarks to the Author):

Flores-Solis et al. report the use of an integrated structural biology approach to investigate the structure and dynamics of a clinically important neoantigenic peptide, NRAS(Q61K), in HLA-A*01:01. It is an excellent manuscript that I thoroughly enjoyed reading. Each of the major conclusions is robustly supported by multiple lines of evidence, e.g., both MD simulations and HDX experiments support differences in the flexibility of NRAS-Q61 peptide variants in HLA-A*01:01. Although I can't comment on the technical aspects of the NMR or HDX experiments, the biochemical experiments and molecular simulations are technically sound. The manuscript is well-written and easy to follow, and the methods are described in an appropriate level of detail.

The integrated structural biology approach outlined here should be widely applicable, and allow analysis of the structure and dynamics of peptide-HLA complexes not amenable to X-ray crystallography, especially moderate-affinity complexes where the conformation of the peptide may not be well-resolved even if crystals can be obtained. The structures of NRAS(Q61K)/HLA-A*01:01 as well as other pHLA complexes obtained using a similar approach will be useful both for basic immunology and immunotherapy development. I think the paper will also be appreciated by the broader structural biology community as an example of integrated structural modeling. I have only a couple of minor concerns about the agreement between the structures obtained by docking and MD simulations, as discussed below.

We thank the reviewer for their positive appraisal of our work, and for helpful suggestions to strengthen our conclusions.

Minor comments

1. The authors investigate the assembly of the NRAS(Q61K)/HLA-A*01:01/b2m complex in significant detail and show that binding involves global structural changes in the HLA-A*01:01/b2m complex. However, the relevance of this analysis was not really clear to me. The kinetics and mechanism of peptide binding are presumably very different for the native HLA-A*01:01/b2m complex *in vivo* and the pH-shocked, refolded HLA *in vitro*. I suggest to provide some additional discussion of the context and implications of these results.

These studies we performed to gain mechanistic insights into the interactions between the MHC-I and NRAS^{Q61K} peptide. Direct peptide interactions with empty MHC-I molecules remain poorly understood but are thought to be a driving factor in antigen selection and pMHC-I cell surface stability. Thus, we sought to characterize how NRAS^{Q61K} could affect the structure and stability of empty MHC-I complex. *In vitro* studies of peptide interactions with recombinant empty MHC-I complexes have been previously reported in the literature and are treated as a first approximation pMHC-I interactions *in vivo*²⁻⁵.

That being said: the reviewer raises an important point. We acknowledge that care must be taken in interpretation of the biological relevance of our *in vitro* results obtained with refolding molecules lacking membrane association and glycosylation, as well as their implications for *in vivo* peptide loading. Evidence from the Elliott lab suggests that glycosylation does not affect peptide binding to MHC class I⁶. However, *in vivo* MHC-I glycosylation could influence interactions with molecular chaperones important for mediating peptide exchange (Tapasin, TAPBPR), and likely plays role in peptide selection⁷. These caveats have been included in the updated manuscript on **lines 163-170 and lines 454-460**.

2. Line 80: "We previously predicted that a public, decameric epitope derived from the recurrent NRAS55-64 mutation, Q61K (ILDTAGKEEY), could interact with the common HLA-A*01:01 molecule." The authors might want to mention explicitly here that the interaction has not only been predicted, but more importantly, experimentally validated in vitro and in vivo.

Agreed. In the revised text we have cited several references on **lines 81-82** that have performed *in vitro* and *in vivo* validation of Q61K peptide binding prior to this study.

3. Line 267: "suggesting limited cross-reactivity of TCRs recognizing NRAS(Q61K) with other known epitopes restricted by HLA-A*01:01". This is a major overstatement that should be deleted. Each HLA presents thousands of potentially cross-reactive antigens - comparison of NRAS(Q61K) with three randomly chosen antigens reveals very little about potential cross-reactivity.

To tone this down, the text on **lines 273-275** has been changed to: "While the conformation of the peptide N- and C-termini was conserved and anchored through interactions within the A/B- and F-pockets of HLA-A*01:01, divergence was observed between residues 4-7 (Fig. 3i), revealing a distinct surface chemistry and geometry for NRAS^{Q61K} relative to other known HLA-A*01:01 epitopes."

4. Line 306: On the significance of differences in deuterium uptake between complexes, could the authors briefly comment (maybe in the methods section) on the criteria for significance? e.g., if only one replicate was done for each complex, how was measurement error estimated.

To address this comment, we performed three new independent recombinant protein refoldings for each protein and then performed HDX-MS in biological triplicates. To determine statistical significance of observed differences, we carried out Brown-Forsythe and Welch ANOVA tests on the mean % deuterium uptake for the representative peptide fragments shown in Fig. 4c and 5e, where $P > 0.12$ (ns), $*P < 0.033$, $**P < 0.002$, and $***P < 0.001$. Updated methods, main text, Fig. 4-5, Supplementary Fig. 8, 11-13, and Supplementary HDX tables 5-8 can be found on **lines 312-323 and lines 1251-1258**.

5. Fig. 6b. This figure shows the representative conformations of each NRAS(55-64)/HLA-

A*01:01 variants in MD simulations. However, it is not clear whether the conformations represent genuine flexibility of the peptide, with multiple transitions between each conformation, or alternatively, relaxation of an incorrect initial conformation into a more favorable conformation. Some kind of analysis of RMSD vs time should be shown to distinguish these possibilities.

Given the observation that the starting pose is a minor conformation in the simulation ensembles, plots of RMSD compared with the starting pose could obscure transitions between distinct states that are similarly distant from the starting pose in RMSD space. Therefore, we have included plots of backbone RMSD vs. Time compared with the centroid of the major backbone conformation for each mutant, and would suggest viewing them alongside RMSF measurements (Figure 6A), backbone dihedral order parameters (Supplementary Figure 14), and χ_1 side chain dihedral angle vs. time plots (Supplementary Figure 15). These plots show that in most cases, flexibility does seem to correspond with back-and-forth transitions in both phi/psi backbone dihedral angles and χ_1 side chain dihedral angles, rather than simply relaxation to a more favorable conformation. A notable exception to the observation of back-and-forth transitions is Q61L, which is stable in 2 out of 3 replicates, but “flips” to an alternative stable backbone conformation during the 3rd replicate, resulting in the observed distribution of backbone poses.

6. Fig. 6b. It appears that the conformation of NRAS(Q61K)/HLA-A*01:01/b2m modeled by NMR and docking represents only a minor conformation (~5%) in the MD simulations. Does the major conformation observed in MD simulations potentially provide a better model of the NRAS(Q61K)/HLA-A*01:01/b2m complex than that obtained by docking? Does this conformation agree with the experimentally determined NOEs? Since docking does not explicitly account for water, it makes sense that MD could give an improved model.

As shown in the new Supplementary Table 4 (lines 340-342), we computed NOE violations for each frame of the NRAS^{Q61K}/HLA-A*01:01 trajectories, including the centroid of the major conformation cluster. This centroid is found to violate 14 of the 25 experimentally determined NOEs, which makes sense given the observed flexibility of the peptide backbone and Q61K sidechain as shown in Figure 6. While we would not necessarily argue that the major conformation is a better model of the complex than generated using NMR and docking, we do think that the ensemble of structural models generated by MD simulation provides a useful tool for understanding the dynamic behavior of the system, and especially the results of our HDX-MS experiments.

7. Line 450, fig. S11. The AlphaFold2 model should also be compared with the poses obtained by MD - it may be that the conformation observed in the AlphaFold2 model is also part of the conformational ensemble. Also, is the AlphaFold2 model consistent with the experimental data? Addressing these two points would make the comparison more persuasive.

We present plots of Q61K backbone RMSD vs. time compared with the AlphaFold2 model's backbone in the revised Supplementary Figure 17. These plots seem to show that the AlphaFold2 backbone model is not found within the conformational space sampled by MD. Additionally, the

AlphaFold2 model violates 16 out of 25 experimentally observed NOEs (inter-hydrogen distance $>6 \text{ \AA}$), which is more violations than the average frame from the MD trajectory and more than the centroid of the major conformation.

8. Lines 464-476: This discussion about the importance of conformational flexibility for TCR recognition is slightly overstated, in my opinion. e.g., line 468 - differential recognition of Q61K and Q61 by TCRs could simply be due to the different chemical structure and properties of Gln vs Lys; there's not necessarily a need to invoke dynamics. Line 474, NRAS Q61K and Q61R are proposed to be especially promising neoantigens for antibody targeting due to a distinct conformation compared with the WT peptide - but presumably many neoantigenic peptides show subtle changes in conformation and dynamics compared with the WT peptide - there is no evidence that Q61K and Q61R are unique in this regard.

We agree with the reviewer's comment and have toned down our discussion accordingly on **lines 491-496.**

Minor corrections

1. The reference numbering is off - e.g., ref. 38 at line 128 and ref. 41 at line 143.

This issue has been fixed in the revised manuscript.

2. Line 347: Q61R is stated to sample three conformations, but according to Fig. 6b there are only two conformations. Maybe the legend in Fig. 6b is mislabeled.

We apologize for any confusion. Fig. 6b shows that Q61R samples two backbone conformations (middle panel), while Fig. 6c shows that of the two backbone conformations, Q61R samples three major side chain conformations (right panel). The text on **lines 358-360** has been revised to make this clear.

3. Line 964 - how was the peptide prepared from the fusion protein?

The method section Protein Expression and Purification has been revised to include more details on the preparation on **lines 889-891 and lines 896-900.**

4. Line 1249 - what model was used for fitting?

A simple single-site (one phase association) model was used for fitting. This has been added to the methods section on **line 1170.**

5. Line 1268 - what modeling was performed, exactly? Was some kind of energy minimization or docking performed at this stage, or was the side chain simply replaced without any change to the backbone?

The side chain was simply replaced without any change to the backbone. Energy minimization was performed during the initial minimization phase of the MD simulations. We have provided this information on lines 1200-1202.

6. Fig. S4a/b - y-axis of two top panels should presumably be "relative absorbance". What wavelength was used?

Yes, relative absorbance at 205 nm. This has been updated in the Fig. S4a/b and in the figure legend.

7. Fig. 7d - I suggest moving this panel to a separate figure in the SI - it's difficult to interpret due to its small size.

We have increased the size of Fig. 7d to be easily interpreted and updated the main text figure.

Reviewer #2 (Remarks to the Author):

Sgourakis and coauthors present an interesting dissection of the binding mode for a neoepitope NRASQ61K to HLA-A*01:01 and the wider HLA allelic landscape, in support of an assessment of this neoepitope's value as an immunotherapy target. This analysis also includes an assessment of how a range of other RAS55-64 neoepitopes bind to HLA-A*01:01. As the authors note, several other studies have defined the TCR repertoires and peptide/HLA-mediated tumor killing associated with NRASQ61K. But they emphasize that studies on the assembly, conformational plasticity, and structure of select NRASQ61K/HLA combinations are lacking. Structural studies have been done for other NRAS neoantigens like RAS5-14, so in this sense the current study completes a key missing piece in the assessment of NRAS for TCR immunotherapy. It does reveal the probable structural uniqueness of NRASQ61K/HLA-A*01:01 and support its use as a target for immunotherapy.

It is a high quality integrative structural modeling study that plugs a hole. The comparative analysis with AF2 Multimer is intriguing, highlighting the ongoing need for structural work in this post-AlphaFold world. The resulting experimentally-generated conformer is sensible and certainly intriguing, but I believe this study belongs in a more specialized journal, given the related body of work that precedes it.

We thank the reviewer for the positive appraisal of our work and comments. Regarding their suggestion for a more specialized journal, we respectfully disagree given the relevance of our study for emerging applications in neoantigen-targeted therapeutics and vaccine design focusing on these public NRAS epitopes, our determination of a challenging and much sought-after pMHC structure, and our investigation of its conformational dynamic properties. We feel that the broad, multi-disciplinary character and significance of our work is appropriate for the readership of *Nature Communications*.

I cannot comment on all the technical elements of the study. I'll restrict detailed comments to the HDX-MS work. Figures 4 and 5 are insufficient to describe the HDX-MS data, and must be expanded.

1. Fig 4c attempts to show a comparative analysis by requiring the reader to compare the "empty" vs NRASQ61K form. Firstly, these should be referred to as the free vs bound samples throughout, and secondly, this is simply too coarse of a comparison to help the reader. Difference plots should be used to support a "delta" deuteration visualization of the datasets.

The empty form has been defined as peptide-free on line 97. A difference plot was incorporated as Supplementary Figure 9 on line 1394, showing the change in HDX at 600 sec between peptide-free and NRAS^{Q61K} bound HLA-A*01:01/h β ₂m protein complexes. The differences in % deuterium uptake between peptide-free and NRAS^{Q61L}-loaded A*01:01/h β ₂m are resolved to individual peptide fragments and mapped on structure (PDB:6MPP) on a white to orange scale. The average % deuterium uptakes were calculated from three independent protein sample preparation and HDX-MS experiments.

2. To support this difference plot, the authors must establish the statistical criteria they used to make the assessment of difference. A simple visual of the two kinetics datasets is insufficient. I strongly recommend they follow a recently-published “guidelines” article published in Nature Methods 3 years ago (Masson et al., Nat. Methods 2019, 16, 595-602).

In the revised manuscript, we have followed the recently published guideline article (Masson et al.) as the reviewer suggested to prepare Supplementary HDX Tables 5-8 and compared the difference in HDX between peptide-free and NRAS^{Q61K}-peptide-loaded HLA-A*01:01 at 600 sec. We further attach a detailed consideration of the guidelines proposed by Masson et al as a supplementary document. We determined that the threshold for significant differences in HDX is 0.297D (Average+1SD). The resulting peptide fragments that demonstrated changes in HDX between peptide-free and NRAS^{Q61K}-peptide-loaded HLA-A*01:01 are located within α_1 and α_{2-1} helices (Supplementary HDX table 8), consistent with our difference plot. In the comparison of representative peptide fragments, we performed the Brown-Forsythe and Welch ANOVA tests, where $P > 0.12$ (ns), $*P < 0.033$, $**P < 0.002$, and $***P < 0.001$. Data collection was performed on peptide-free (empty), NRAS^{Q61K}, NRAS^{Q61}, NRAS^{Q61H}, NRAS^{Q61L}, and NRAS^{Q61R} peptide-loaded HLA-A*01:01/h β_2 m proteins in biological triplicates (three independent *in vitro* protein refolding). Lines 1240-1258 of the main text have been modified to reflect these points.

3. These guidelines also contain information on supplementary datasets that should be provided for the reader to gauge the completeness of the study. Note in particular that the mapping of deuteration changes (or kinetics) should only be based on the peptides that survive deuteration analysis, and not simply the sequence map of the undeuterated data.

We apologize for the confusion and missing supplementary information. We have provided Supplementary HDX Table 5 to summarize the HDX experiments and Supplementary HDX Tables 6-7 to emphasize that the measurement of deuterium uptake is resolved to individual peptide fragments in the method session on lines 1240-1258.

4. Figure 4a shows a peptide that appears to either demonstrate EX1/EXX kinetics (139-154), overlap, or partial binding. It adds concern over the way data analysis was conducted, as this peptide is indicated to demonstrate a large difference in deuteration upon neoepitope binding (Fig 4b). As the guidelines will indicate, a clearer statement on selection criteria may help, in addition to the provision of all kinetics curves if possible.

We thank the reviewer for highlighting Fig. 4a, lines 281-284 and lines 287-290. We have modified the main text to clarify that “Compared to control experiments performed in the absence of deuterium, HDX performed on empty and NRAS^{Q61K}-bound HLA-A*01:01/h β_2 m samples for 600 seconds resulted in different uptake of deuterium for peptide fragments spanning the entire HLA-A*01:01 and h β_2 m molecules,” which is consistent with Fig. 4b. Selected peptide fragments

reflect the significant difference in HDX within α_1 and α_{2-1} helices between the peptide-free and the NRAS^{Q61K}-peptide-loaded HLA-A*02:01 (delta HDX>0.297, Supplementary HDX Table 8).

5. In connection with this concern, I cannot see what concentration excess of neoepitope to HLA was used in the HDX study. This should be provided as I'm sure it was done to saturate binding (note that upon deuteration there is a high dilution, requiring concentration excess in most cases). And if there was a large excess, how did the authors manage to detect changes in the NRAS peptides themselves? Information is missing.

No excess neoepitope peptide was used for the HDX studies on peptide/MHC-I complexes. All pMHC-I complexes were prepared using *in vitro* refolding. The heavy chain HLA-A*01:01 and light chain β_2m are refolded *in vitro* in excess peptide and purified by size exclusion chromatography, resulting in 1:1:1 stoichiometric complexes. The resulting protein was immediately flash-frozen and freshly thawed for each HDX-MS experiment to avoid peptide dissociation. We do not measure HDX-MS of the titration of mutant peptides for binding but the peptide-free or NRAS^{Q61K}, NRAS^{Q61}, NRAS^{Q61H}, NRAS^{Q61L}, and NRAS^{Q61R} peptide-loaded HLA-A*01:01/h β_2m in deuterated buffer as protein complexes. This information is included on lines 310-312 (main text) and 1227 (methods).

Reviewer #3 (Remarks to the Author):

Flores-Solis and coworkers combine different biophysical (most notably NMR) and computational methods (Rosetta docking and MD simulations) to determine the solution-state structure of a peptide/human leucocyte antigen (pHLA) complex, NRASQ61K/HLA-A*01:01. This combined integrative computational structural biology approach enabled the authors to determine an atomic-resolution conformational ensemble from a (relatively sparse) set of 25 NOE distance constraints between the peptide and the HLA protein (in addition to a measured dense network of NOE contacts within the HLA protein itself). Using MD simulations, the authors extend their study also to other RAS neopeptides, such as Q61L, Q61R, and Q61H. Finally, using in-silico and in-vitro approaches, they also examined the HLA binding repertoire of RAS neopeptides beyond HLA-A*01:01 towards the HLA-B*15:01 and HLA-C*08:02 allotypes.

The amount of work done and the results obtained are impressive. The experiments and computations were carefully done and interpreted, and the main conclusions are supported by the data. Concerning the computations, established state of the art methods were used. However, we have a few requests for further improvements and mutual validation of MD and NMR.

We thank the reviewer for their positive comments on our work, and its significance. Please see detailed responses to the very insightful comments and suggestions below.

1) The available NMR distance constraints provide a great opportunity for the authors to more deeply scrutinize and carefully validate their MD simulations against their NMR data, by quantifying the NOE restraint violations in their MD simulations of NRAS-Q61K. This analysis is straightforward (for example with gmx disse, in which also proper $1/r^6$ distance averaging for NOE distances is implemented, but it can of course also readily be done differently) and will provide insights into the mutual consistency of MD and NMR. In addition to the overall average violations (global picture), it will be insightful to see which of the 25 NOE-derived distance constraints are violated (if any), by how much (assuming a reasonable upper distance bound of, e.g., 6 Å), and maybe even obtain insights into the reasons underlying the deviations.

We thank the reviewers for this suggestion, and have performed the suggested analysis, with results in Supplementary Table 4 (lines 340-342). The table shows that the least-frequently-violated NOE restraints correspond with the anchor residues (L56 and Y64), while contacts near the center of the peptide are frequently violated in simulation. This makes sense given the flexibility of the Q61K peptide backbone as shown in Figure 6.

2) Which of the 25 NOE contacts are essential for determining the conformation ensemble of the complex, and which ones are not so important? For example, would a few constraints at the N- and C-terminal ends of the peptide (pockets A/B and E/F, respectively) suffice to obtain a similar, well-defined structural ensemble? Using only subsets of the NMR distance constraints for the modeling also enables the authors to use the violations of the unused NOE

contacts for cross-validation.

The reviewer brings up an interesting suggestion. During initial structure calculations we played with including subsets of NOEs and observed how that changed the convergence of the peptide ensemble. Without any NOE restraints (Fig. 3b-e), docking does not perform very well at all, and the peptide conformation is completely wrong. When ~1/3rd of the NOEs are left out we obtain a correct "general" binding mode (*i.e.*, directionality of N- to C-terminal binding), but the peptide convergence is poor. Thus, we found that a major of the NOE network was necessary to get good structural convergence in our final ensemble, likely because the data are sparse.

MINOR ISSUES:

1) p. 3, line 133: It should be Ref 37 here, not Ref 40.

This issue has been fixed in the revised manuscript.

2) p. 7, line 321: We would not necessarily agree that the solvent accessibilities are unique. For example, different hydrophobic side chains at position 61 might have very similar accessibilities. What the authors mean there is probably that the solvent accessibilities of the different Q61 variants are DIFFERENT (but not necessarily "unique" in a strict sense).

We have modified the word "unique" to "different" on line 331.

3) p. 8, line 338: The authors stress the agreement between the S2 order parameters of backbone phi/psi dihedrals and the RMS fluctuations of the backbone atoms. However, this "agreement" is trivial because the backbone RMSF and phi/psi order parameters contain highly redundant information. Backbone atomic fluctuations are largely due to phi/psi dihedral fluctuations, as these are the two most flexible degrees of freedom of the peptide backbone. So the agreement is obvious and should not be overly stressed.

We thank the reviewers for this comment, and agree that the information contained in RMSF and phi/psi dihedral order parameters is largely redundant. We have revised the text to reflect this on line 350-352.

4) p. 8, line 347: The authors discuss the "reduction of conformational space sampled by the Q61L peptide, which can be described by only two clusters relative to the other peptides (3, 3, 4, and 5 for Q61, Q61R, Q61H, and Q61K, respectively) (Fig. 6b,c)." We think this is unclear from Fig. 6b,c because for Q61 and Q61R, for example, there are also only 2 backbone clusters -- but we assume the authors here refer to the larger number of side chain clusters than for Q61L? Could this be clarified in the text, to avoid misunderstandings?

We appreciate the reviewer's comment, and have revised the text to clarify that the "reduction of conformational space" refers to the smaller number of side chain clusters observed for Q61L in comparison with other amino acids at that site on line 358-360.

5) p. 8, line 353ff: Are there data to support the reduced entropy loss and/or more favourable binding free energy? If not, these aspects should be clearly marked as speculation.

We thank the reviewers for this comment; we have revised the text to clearly label these claims as speculative on **lines 365-366**.

6) p. 37, line 1263: The sentence "Periodic cubic boundaries were used" sounds odd. We guess what the authors mean here is that the simulations were carried out in cubic simulation boxes, and that periodic boundary conditions were used in all three spatial dimensions?

We thank the reviewers for this comment; we have revised the text to use the suggested language to describe our simulation methodology on **lines 1193-1196**.

7) p. 37, line 1264: Can the authors specify what they exactly mean with "V-rescale modified thermostat"? We guess they refer to the velocity rescaling thermostat with a stochastic term by Bussi et al, which is applied via the "v-rescale" mdp option in Gromacs? If yes, please rewrite accordingly and add a reference to Bussi et al.

We thank the reviewers for this comment; we were indeed referring to the velocity rescaling thermostat applied using the v-rescale mdp option, and have clarified the text to show this. As requested, the Bussi et al reference has also been added on **lines 1193-1196**.

Reviewer #4 (Remarks to the Author):

In this manuscript by Flores-Solis et al. the properties of MHC class I (MHCI) molecules in complex with a neoantigen from Ras (the NRAS peptide NRASQ61K) is investigated by biophysical and structural means. This question is of importance, since it has become clear that the conformational dynamics either of pMHCI or of the TCR are of critical importance for understanding TCR specificity and affinity. For example, a recent paper in Nature Communications was good evidence for this hypothesis. While the crystal structure of a lower affine TCR and its high affine engineered counterpart showed very little difference in the mode of binding and could not explain affinity enhancement, thermodynamic analysis and MD simulations revealed electrostatic and entropic contributions to the binding free energy (Poole et al., Nat Commun 5333, 2022). In this manuscript, the characterization of the complex is vigorous, employing several biophysical techniques. Empty MHCI is analyzed as well as the neoantigen bound complex. In particular NMR spectroscopy is used to delineate conformational dynamics. Line broadening is severe for the empty MHCI, and as expected spectra become much more disperse and amenable to analysis once peptide is bound. NOE analysis reveals the key interactions in the binding groove and allows to interpret chemical shift changes in the structural context. Moreover, HD exchange in combination with mass spectrometry is utilized to derive the stability of individual secondary structure elements. It becomes evident, that the empty MHCI is largely destabilized. However, two of the underlying b-strands are stabilized compared to the peptide-bound MHC, indicating that residual stable secondary structure is maintained. Furthermore, different mutants of the NRAS peptide (Q61K, Q61R, Q61H, and Q61L) were probed by the HDX method and complemented by MD simulations. One mutant, Q61L showed decreased solvent accessibility (HDX-derived) and more restricted motions (MD). In contrast the Q61R and A61K mutants show the charged side-chains to be fully exposed and thus represent critical hot spots for the design of engineered TCR's or antibodies.

This is an important manuscript in the area of neoantigen presentation by MHC class I molecules. However, there are several issues that should be addressed prior to publication:

We thank the reviewer for their comments on the significance and importance of our study, and for pointing out the previous work that motivated our studies. Please see detailed responses to the very insightful comments and suggestions below.

1.) According to the HSQC in Figure 1d, roughly 50% of the MHC is receptive, while the other 50 % need to equilibrate prior to binding the peptide. In contrast in Fig. 1g-i, where the peptide resonances are recorded, no such biphasic behavior is observed. How do the authors explain this discrepancy ? Also, can they say more about the two conformations, e.g. based on their H/D experiments ? Are oligomers of the empty MHCI formed ?

The reviewer is correct, we are observing bi-phasic behavior only from the perspective of the labeled MHC-I molecule as opposed to the peptide. This is likely due to the fact that, unlike the MHC-I, the peptide does not require a large conformational change in order to bind to the MHC-

I. On the basis of our NMR and size exclusion chromatography data we do not observe a significant amount of empty MHC-I oligomers or other higher order species, but instead a slow conformational transition between monomers.

2.) Structural analysis has been performed for the peptide-bound MHCI, but I was wondering whether NOESY spectra were also recorded for the empty MHC molecule and whether this would indicate in more detail which regions are destabilized or stabilized. The H/D exchange data shown later indicate that while most of the binding groove is destabilized, residues 22-30 for example are stabilized. NOE's should be observable for those residues in contact with the neighboring strands. This point relates to the question also asked in 1.) and which concerns the difference between the non-receptive and the receptive MHCI. What are the stabilizing interactions that make the energy barriers towards the receptive conformation so slow ?

The reviewer makes a good point. However, collection of NOESY spectra for empty MHC-I is not possible due to 1) an inability to prepare sufficiently concentrated samples for collection of 3D experiments, and 2) observations that NMR resonances corresponding to residues in the antigen binding groove are completely broadened in the empty state, and thus would not provide NOE cross-peaks.

3.) H/D exchange is combined with mass spectrometry, yielding very good sequence coverage. What is missing, however, is a Table that clearly denotes the (i) relative deuterium uptake at (ii) what time point and (iii) for which sequence.

We thank the reviewer for pointing out the missing supplementary information. We have included **Supplementary HDX Tables 5-8**, according to community-established guidelines (Masson et al., Nat. Methods 2019, 16, 595-602), which contains raw deuterium uptakes and back-exchange corrected % deuterium uptakes at all six time points of 0, 20, 60, 180, and 600 sec for individual peptide fragments done in biological triplicates.

4.) When discussing the different HLA alleles, HLA-A*01:191 is pointed out as a high affinity binder, potentially cross-reactive with HLA-A*01:01. It would be important that the authors verify their predictions by experimental evidence, express the A*01:191 allele and measure stability/exchange and to then directly compare, e.g. thermal stabilities. Combining this with for example peptide NMR measurements would indicate whether the observed kinetics of peptide binding is similar for two allotypes, thus allowing to conclude whether the slow conversion of a peptide-free to a peptide-receptive MHC is a conserved phenomenon.

We thank the reviewer for highlighting the importance of experimental validation. Therefore, we performed *in vitro* refolding of HLA-A*01:191 with β_2m in the presence of NRAS^{Q61K} peptide. We obtained a good yield of refolded protein complex consistent with computational prediction, shown by the size exclusion chromatography. Additionally, DSF analysis demonstrates that the loading of NRAS^{Q61K} peptide results in a ternary HLA protein complex with a T_m of 58.5°C, higher than the T_m (52.5°C) of NRAS^{Q61K}-loaded HLA-A*01:01. We have added the Supplementary Fig. 16 and added **lines 392-395** to reflect these new results.

5.) From the further analysis of B*15:01 and C*08:02 alleles it is found that they show very similar stabilities (and thus affinities) in complex with the NRAS mutagenic peptide as the A*01:01 allotype. However, predicted binding affinities are lower. How can this be explained and would it not raise the question how reliable the predictions are?

First, while DSF experiments suggest NRAS^{Q61K}/A*01, NRAS^{Q61K}/B*15, and NRAS^{Q61K}/C*08 exhibit similar thermal stabilities, they may exhibit unique kinetic stabilities, as suggested by quite different yields obtained from in vitro refolding experiments. The reviewer is correct that these results are inconsistent to binding predictions that are known to have limited quantitative agreement with the experimental T_m measurements.

6.) In perspective, the authors make the claim that antibodies targeting Q61K or Q61R would be a good choice for tumor targeting. However, this is somewhat self-evident given the site of mutation. More interesting is the remark that the dynamic features and conformational differences of the peptides could be exploited by engineering TCR's. Would the authors have a concrete suggestion of how this engineering would profit from their data ? A crystal structure of the TCR bound to the pMHCI might also not solve the issue (see paper by Poole et al., Nat. Commun. 2022), so the dynamic and entropic implications of binding might be critical. A matched pMHCI-TCR tri-molecular modeling approach might therefore be the way to go, so I was wondering about the authors point-of-view regarding this aspect.

We thank the reviewer for their very insightful comment. We agree that our observation of unique dynamic features for the different NRAS neoantigen mutants can be exploited for engineering TCRs, through different avenues. First, given that TCR binding to form a high-affinity signaling complex likely necessitates some degree of peptide conformational adaptation of the peptide, it is plausible that minor conformations sampled by each neoantigen can be targeted using conformation-selective receptors. Second, due to the contribution of the loss of conformational entropy upon binding to the overall binding free energy, more rigid peptides could potentially present a better epitope for molecular recognition by engineered TCRs. Lastly, as suggested by the reviewer, our structures can be exploited for modeling complexes with existing, cross-reacting TCRs that can be further redesigned to accommodate each mutant. This is an approach that we are currently following in our group.

Reviewer #5 (Remarks to the Author):

In this manuscript, Floris-Solis et al. describe the structural and dynamic properties of NRAS variants bound to HLA-A0101. The RAS epitopes are of significant interest as they are immunotherapy targets, with potential utilization in cancer vaccine or T cell therapy approaches. The manuscript takes an interesting approach in using nuclear magnetic resonance (NMR) as opposed to crystallography to characterize the Q61K NRAS neoantigen/A0101, followed by hydrogen deuterium exchange, modeling, and simulation to look at how other Q61 variants might lead to altered structures and dynamics.

Major strengths of the manuscript are the high-quality work that is involved and the significant attention to detail. The authors really should be commended for their comprehensive efforts and beautiful presentation of the data. The downside of the paper, however, is that the manuscript is somewhat of a grab-bag of protein biophysics/structural immunology that **generally** confirms much of what is already known, as detailed/demonstrated by NRAS peptides, without new advances for structural immunology or RAS-connected immunology/immunotherapy.

For example, the manuscript opens with a demonstration that peptide loading follows a slow conformational transition from empty HLA to peptide bound. This has been well known for quite some time. Yes, there is more detail here, and it is nice to see it done with an NRAS peptide, but not much new is learned. Later, we go to findings that different NRAS variants result in different hydrogen-deuterium exchange and (via modeling) different dynamic envelopes. Again, it is already known that small changes to peptides can lead to different HLA and peptide motions and that these can be impactful for immune recognition. While the details to the NRAS peptides are new, there is no reason to suspect this system would be an exception. Because there is no connection to T cell (or TcR) recognition, peptide potency, vaccination, immunotherapy etc., there is little that is actionable. Rather, what we have is a beautiful biophysical study that confirms what is generally well known. Of course, the use of NMR and H-D exchange, structural modeling, etc. is relatively new to the field, but outside of its impressive technical aspects, I found the paper to be of low potential impact.

We thank the reviewer for their overall positive appraisal and comments on the high quality of our work. Regarding their comment on impact, we respectfully disagree given the relevance of our study for emerging applications in neoantigen-targeted therapeutics and vaccine design focusing on these public NRAS epitopes, our determination of a challenging and much sought-after pMHC structure, and our investigation of its conformational dynamic properties using a novel, integrative pipeline encompassing complementary biophysical techniques. We feel that the broad, multi-disciplinary character and significance of our work is appropriate for the readership of *Nature Communications*.

Some specific comments:

1) I missed the structural validation of the other variants at position 61. With NMR

determination of the K variant, how were the others determined? I think it is all in silico replacement and modeling? This would seem to place a question mark over the interpretations.

In vitro characterization of other NRAS Q61 variants was performed using complementary experimental (DSF, FP, HDX-MS) and computational (MD, Rosetta) approaches. With regard to structural determination, the models used in MD simulation were generated by threading the peptide side chains without changing the backbone conformation, and minimization was performed as part of system preparation for MD simulation. While we did not determine experimental structures for variants other than NRAS Q61K, we believe that our structural models and their associated MD structural ensembles are consistent with and helpful in interpreting experimental data such as the HDX-MS results.

2) The argument that peptide dissociation/exchange prevented crystallization seemed handwaving to me. Certainly lower stability complexes have been crystallized. This seemed like an argument to justify the use of NMR in this case.

We agree with the reviewer that a limited number of MHC-I complexes with lower affinity peptides have been crystallized in the past. However, for other important antigens crystallography has not produced high-resolution structures, (see for example Ebrahimi-Nik et al., Nat Commun, 2021) and therefore structural modeling is being used instead. The purpose of our NMR-based, integrative modeling approach is not to replace X-ray crystallography, but to demonstrate a facile alternative that can deliver high-resolution structural information. We have revised the manuscript on **lines 463-466** to include this important point.

3) Is it really necessary to show protein production, etc. as supplementary information? I found that additional detail is distracting, along with the segue into the protein conformational change. The overall story could use some editing to get to a main point.

Given that this manuscript provides the first biophysical/structural characterization of NRASQ61K/HLA-A*01:01, and that preparation of this complex is essential for the later biophysical analysis, we believe it is necessary to provide information and details of its production and purification in the supplement, to ensure the reproducibility of our work by other labs. We note that this is a very small part of the entire manuscript so it shouldn't detract from its primary focus.

References

1. Perez-Riverol, Y. *et al.* The PRIDE database resources in 2022: a hub for mass spectrometry-based proteomics evidences. *Nucleic Acids Res.* **50**, D543–D552 (2021).
2. Eisen, H. N. *et al.* Promiscuous binding of extracellular peptides to cell surface class I MHC protein. *Proc. Natl. Acad. Sci.* **109**, 4580–4585 (2012).
3. Saini, S. K. *et al.* Empty peptide-receptive MHC class I molecules for efficient detection of antigen-specific T cells. *Sci. Immunol.* **4**, (2019).
4. Anjanappa, R. *et al.* Structures of peptide-free and partially loaded MHC class I molecules reveal mechanisms of peptide selection. *Nat. Commun.* **11**, 1314 (2020).
5. Sun, Y. *et al.* Universal open MHC-I molecules for rapid peptide loading and enhanced complex stability across HLA allotypes. *Proc. Natl. Acad. Sci.* **120**, e2304055120 (2023).
6. Haurum, J. S. *et al.* Recognition of carbohydrate by major histocompatibility complex class I-restricted, glycopeptide-specific cytotoxic T lymphocytes. *J. Exp. Med.* **180**, 739–744 (1994).
7. Ilca, F. T. & Boyle, L. H. The glycosylation status of MHC class I molecules impacts their interactions with TAPBPR. *Mol. Immunol.* **139**, 168–176 (2021).
8. Mayne, L. *et al.* Many Overlapping Peptides for Protein Hydrogen Exchange Experiments by the Fragment Separation-Mass Spectrometry Method. *J. Am. Soc. Mass Spectrom.* **22**, 1898–1905 (2011).

Detailed answers to Mason et al. sample preparation and analysis recommendations

11.

We recommend that a sample quality assessment precede the HDX experiment. The assessment could include denaturing electrophoresis (SDS-PAGE) analysis and intact protein mass spectra to confirm sample purity and confirm the expected sequence and post-translational modifications. A size-exclusion chromatography or native MS analysis is useful to establish the monomeric or oligomeric state of the sample being investigated, or a functional biochemical assay to check that the protein is active and correctly folded. We recommend a size-exclusion chromatography analysis to establish the monomeric/oligomeric state of the sample being investigated.

Detailed characterization of recombinant pMHC-I complexes utilized for HDX studies is presented in Supplementary Figure 2. Size-exclusion chromatography, SDS-PAGE analysis, LC-MS analysis confirm the identity of expected components, molecular weights, and reveal a monomeric state of pMHC-I complexes.

13.

Sample preparation is key to a reliable HDX-MS experiment. Given the sensitivity of HDX-MS, any perturbations in pH (or pD), temperature and ionic strength will have considerable effects on the outcome of any experiment, and so it is crucial to control these parameters. At a minimum, the buffer used in the labeling reaction must have sufficient buffering capacity to ensure a constant pH, and the temperature of the labeling reaction must be well controlled. Therefore, the composition of the buffer used in the labeling reaction and the temperature and pHread (pH meter reading with no isotope corrections applied) at which the reaction was conducted must be reported. Both labeling buffer and protein solution must be pre-equilibrated at the temperature of the ensuing HDX experiment and stably maintained during labeling.

Sodium phosphate buffer was used in HDX-MS experiments performed at pH/pD 6.5. The buffering range of phosphate buffer is ~5.8 to 7.4, which is suitable to maintain pH/pD in our studies. All solutions were pre-equilibrated in controlled heat blocks or water baths at 25°C, and all HDX-MS reactions were performed at 25°C (temperature was constantly monitored with thermometers).

14.

The concentration of D₂O (% v/v) present during the labeling reaction must be precisely maintained and clearly reported. HDX experiments may be conducted with any concentration of D₂O, but experiments are typically conducted at higher concentrations of D₂O (80–90%), as this leads to greater deuterium incorporation (resulting in a larger mass shift). Achieving the highest signal-to-noise ratio (that is, sensitivity to distinguish differences in HDX between protein states) and minimizing spectral complexity can require optimizing the concentration of D₂O used, which should be precisely maintained throughout the experiments and reported.

5 µL of 30 µM stock pMHC-I proteins was diluted to 6 µM in HDX Buffer 2, resulting in a final concentration of 80% D₂O (v/v), consistently maintained throughout all experiments.

15.

Quench buffer composition greatly affects the efficiency of digestion, and thus the protocol used for quenching should be reported (composition and pH of the quench buffer). The final composition of the quenched sample, that is, the concentration of labeled protein and quench solutions, as well as the pH of the quenched sample, should also be reported.

All buffers used are reported in the methods section. The buffers are:
HDX Buffer (50 mM NaCl, 20 mM sodium phosphate, pH 6.5 in H₂O or pD 6.5 in D₂O).
HDX Quench Buffer (50 mM NaCl, 1M TCEP (tris(2-carboxyethyl)phosphine), 20 mM sodium phosphate pH 2.35 in H₂O).

The final composition of the quenched sample contains 3 μM of labeled protein.

16.

Repeated measurements of deuterium incorporation are necessary to ensure repeatability and deliver an estimate of the precision in the measurements. Independently generated exchange reactions serve as technical replicates. The same labeling reaction aliquoted and measured separately is not a suitable technical replicate, as this is not an independent observation. At a minimum, there should be at least three labeling reaction experiments performed for at least one time point to allow a reasonable estimate of the error of measured deuterium levels. This estimate of error should be used to support the assignment of significance to differences in HDX between states. Labeling reactions (more than six) performed with extraction of many samples across a wide time range (more than four orders of magnitude) can provide additional confidence in the assignment of statistically significant differences in HDX. Where practical issues arise (for example, restricted sample supply), replicate measurement may not be necessary for all the time points in the reported HDX curves.

Three independent HDX reactions were performed on three independent preparations of recombinant proteins for each sample, serving as proper technical replicates. For each replicate, we acquired four time points across the time range of 0 to 600 seconds to confirm the HDX measurements' reproducibility and consistency and determine the statistical significance between samples.

17.

Biological replicates of the experiment should be conducted where possible. This would require additional preparations of the protein. The repeats ensure that the variability in exchange measurements that can be ascribed to post-translational modifications/differences in protein expression/purification or variable stoichiometry in reconstituted protein complexes is quantified. Biological replicates are especially important for proteins that require extensive sample preparation before HDX-MS (for example, nucleotide loading in monomeric GTPases⁵²).

Our replicates are indeed true biological replicates. For our three HDX-MS replicates, separate preparations of recombinant proteins were carried out and HDX measured independently.

18.

The LC-MS system used to collect the data should be made explicit in the methods section of the manuscript, along with pertinent instrument settings (for example, the LC gradient and flow rates, reversed-phase columns used, MS ion source parameters and so on).

LC separation was performed using a C8 5 μm column (TARGA, Higgins Analytical, TP-M501-C085), and MS/MS measurement were performed on a Q Exactive Orbitrap Mass Spectrometer (Thermo Fisher Scientific).

The details of our instrument set up have been outlined in detail previously (Mayne et al. J Am Soc Mass Spectrom. 2011⁸). The relevant section of the manuscript is shown below:

“Experimental samples are injected into an on-line flow system. The flow system carries the protein solution at 50–120 $\mu\text{L}/\text{min}$ through an immobilized acid protease column for proteolysis (15 or 60 μL total volume each). Digested peptides are directed through a second valve to a small C8 column (1 \times 5 mm, 5 μm beads). After sufficient flow (\sim 3 min at 50 $\mu\text{L}/\text{min}$) to transport the peptides into the trap column and wash away buffer salts, the second valve is switched, placing the trap column in the flow of a low volume HPLC pump. A water/acetonitrile gradient (6 $\mu\text{L}/\text{min}$, 10%–50% AcCN over 10–15 min) elutes peptides from the trap column through an analytical C18 column (0.3 \times 50 mm, 3 μm beads) for rough peptide separation, and then to the electrospray needle for further separation of the peptides by mass. Narrow tubing (25 or 65 μm i.d.) is used in the slow flow chromatography stage. To minimize eluant peptide overlap, we use an acetonitrile gradient shaped to elute constant numbers of peptides per unit time. Typical peptide elution peak widths are 20 sec wide at baseline. We have not found peak widths to be limiting in the MS data even for hundreds of peptides, due especially to the ability of the ExMS analysis program [46] to recognize peptides in MS data even in the presence of significant spectral overlap.

Cleaning steps between serial experiments (several up–down gradients) elute very large peptides not useful for HX-MS experiments, helping to avoid peptide carryover from earlier MS runs and to maintain low column back pressure. If some peptides present a particularly difficult carryover problem [51], the availability of many other peptides makes it feasible to simply remove them from the experimental list. Overall, each experimental cycle takes 20–25 min followed by a 10–15 min cleaning cycle, resulting in a 40 min total time for each run. This allows for as many as 10 experimental HX runs each day in addition to an all-H run to calibrate chromatographic retention time for each peptide and an all-D run to calibrate their back exchange.

The flow rates and column sizes noted were chosen as a compromise between the competing demands of transit time, resolution, and back pressure. Particular care must be taken with flow system connections to assure against leaks under significant back pressure and to minimize unswept dead volume, which can lead to peak trailing and problems with peptide identification, although the ExMS analysis program operates to minimize this problem. Tubing is carefully cut, inspected, and connected, and ports are sprayed out with clean pressurized air on assembly. Common problems include slippage of connectors due to under-tightening, blockage due to over-tightening, and the trapping of particles at connections.”

19. As a variety of proteases are available for sample workup, the protease used in the experiment should be stated. Additionally, the duration of the digestion, the digestion mode (off-line or on-line) and the temperature that the digestion was conducted under should be reported. In the case of on-line digestion, column dimensions, source and flow rate should be specified.

Pepsin digestion was performed on-line (i.e., on the column). pMHC solutions were carried at 50–120 $\mu\text{L}/\text{min}$ through an immobilized acid protease column for proteolysis. Protease columns used here are 1 or 2 mm \times 20 mm guard columns packed with POROS AL to which either pepsin coupled using Na_2SO_4 for salting out the protein during the coupling reaction at room temperature.

20.

Evidence of LC-MS system suitability for reproducible deuterium recovery is required. The level of back-exchange (loss of deuterium) of the particular HDX-MS system and workflow being used must be characterized in detail by analysis of a mixture of model peptides (for example, bradykinin or angiotensin II) or ideally a digest of a model protein (for example, hemoglobin, phosphorylase B or cytochrome c) that has been equilibrated in deuterated buffer for an extended time period (for example, 12 h) to allow complete labeling of all backbone amide NHs in the model

peptide/protein. If such a characterization of the workflow and LC-MS system in use has been performed in an earlier study, that work can be referenced. The characterization serves to validate the workflow and LC-MS setup and ensures that a suitable level of deuterium is retained during any given experiment. The back-exchange of peptides of the model system used can be calculated using the following equation:

$$\text{Backexchange} = (1 - m_{100\%} - m_{0\%} N \times D_{\text{frac}}) \times 100$$

where $m_{0\%}$ is the non-deuterated peptide centroid mass, $m_{100\%}$ is the maximally labeled peptide centroid mass, N is the theoretical number of backbone amides in the peptide and D_{frac} is the fraction of D/H in the labeling buffer used (for example, 0.80, 0.90, 0.95). Back-exchange levels are ideally reported on a per-peptide basis but may be reported as the average percentile loss of deuterium of all peptides analyzed with an indication of the range of values observed, for example, 40% (ranging from 10% to 55%). In a well-conducted conventional state-of-the-art bottom-up HDX-MS workflow, only very few peptides should exhibit back-exchange values above 50%.

Back-exchange reactions were carried out for each sample measured. Details are described below (see 21. below).

21.

We recommend producing a ‘maximally labeled’ control sample of the protein studied (also known as a 100% exchange control), particularly in situations where the absolute amount of exchange is desired. This control allows for an estimate of the level of back-exchange for each analyzed peptide during sample work-up and analysis. A ‘quench exchange’ control sample may also be used to estimate the amount of on-exchange that occurs during the quench process, but unless long quench/digestion times (> min) or non-ideal quench conditions (pH > 2.5) are required, this is usually well-approximated by the non-deuterated centroid mass in a typical bottom-up LC-MS experiment. From this information, a back-exchange corrected fractional deuterium level (D_{corr}) can be estimated using the following equation:

$$D_{\text{corr}} = (m - m_{0\%}) / (m_{100\%} - m_{0\%})$$

where m is the observed peptide centroid mass at a given time-point, $m_{0\%}$ is the non-deuterated peptide centroid mass and $m_{100\%}$ is the maximally labeled peptide centroid mass. When D_{corr} is expressed as a percentage value, it is sometimes referred to simply as the ‘back-exchange’ corrected percentage D value for a given analyzed peptide. A more explicit treatment can be found in ref. 53.

The absolute amount of exchange (D_{absolute}) in the peptide can then be calculated based on D_{corr} as $D_{\text{absolute}} = D_{\text{corr}} \times N$, where N is the theoretical number of backbone amides in the peptide. The absolute amount of exchange can be required for interpreting HDX-MS data in a structural context, for example, for identifying intrinsically disordered regions. These controls are not strictly necessary for comparative measurements between different states of the same protein (for example, with and without ligand), as the back-exchange can reasonably be expected to be the same with each measurement. Furthermore, we recommend performing the labeling reaction that produces the maximally labeled control sample for 12–24 h at room temperature and low pH ($2.5 < \text{pH} < 4$) in the presence of a strong denaturant (for example, 6 M GndDCI or 6 M urea). Such a treatment usually offers adequate HDX equilibration between protein and labeling solution ensuring complete labeling of all backbone amide NHs. However, in rare cases, a maximally labeled control prepared in this manner can fail to exchange a minor subpopulation of very slow exchanging amide NHs, and furthermore sample aggregation can also be a concern for some proteins²¹. Preparation of maximally labeled control samples at higher pH values (pH > 5) and

excessive labeling times (>24 h) or elevated temperatures (>25 °C) should be approached with caution, as histidine residues may, at these conditions, begin to incorporate a substantial amount of deuterium at the imidazole sidechain (c-2 position), leading to higher than expected levels of deuterium⁵⁴. In any case, a carefully prepared, maximally labeled control sample of the target protein under study will serve as the best possible estimate of the maximal level of deuterium one can expect to detect in each peptide analyzed during the given HDX-MS experiment.

Maximally labeled samples for each protein, where samples were heated to 46°C or 35°C for 15 minutes (10 degrees lower than the melting temperature measured for peptide-loaded or empty protein), were used to estimate the maximal level of deuterium, one can expect to detect in each peptide analyzed during the HDX-MS experiment.

22.

A wide range of D₂O labeling times should be used to interrogate the full range of possible amide exchange rates³⁹. We recommend that labeling times span at least four orders of magnitude (for example, 0.1 min, 1 min, 10 min, 100 min and 1,000 min), with the shorter times in the range of 5–15 s, and the longer times lasting at least several hours. Importantly, sampling labeling times beyond four orders of magnitude can provide additional useful information on HDX kinetics and error (see also recommendation 1.5). The exact time range will depend on the protein system in question. For example, experiments on intrinsically disordered proteins, which are likely to exhibit rapid HDX rates, can focus on even shorter labeling times (that is, 0.1 s, 1 s, 10 s or 100 s). For very stable proteins that may have extensive regions that undergo very slow HDX, experiments should be performed with very long labeling times (>1,000 min or 16 h). The selection of time points should be justified in the manuscript; it is important that the time points cover a range sufficient to allow for a substantial change in the deuterium level of the majority of the amides in a protein, while still allowing for the detection of fast exchange events. In short, the range should be targeted to the nature of the investigation, such as shorter labeling times with narrower time ranges for investigating transient molecular interactions or disordered proteins. Selecting the labeling times is particularly important when designing appropriate null hypothesis experiments for differential HDX studies (that is, failing to disprove that there is no difference in HDX between two conditions).

Due to the fast rate of exchange observed for certain regions of the protein, we select ~0.33 min, 1 min, 3 min, and 10 min covering two orders of magnitude. Additionally, we performed triplicates on each time point to estimate error.

23.

Especially for longer labeling times (for example, over 100 min), we recommend conducting a quality control ‘deuterium-pulse’ experiment if the physical stability of the protein of study is unknown under labeling conditions. In such an experiment, the protein is incubated for a time equal to that of the longest labeling time but in the absence of deuterium and is then deuterium-labeled for a short time period (for example, 10–30 s). This pulse-labeled sample should then be compared to the sample from the equivalent short-labeling experiment but without the prior incubation. The purpose of this comparison is to check whether the protein is undergoing structural changes (for example, precipitation, oligomerization or irreversible unfolding) over the course of the labeling time, which may result in misinterpretation of data.

Our longest exchange time was 600 sec. Thus, the suggestion for “longer-labeling times” does not apply to our study.

24.

To ensure optimal sensitivity for detecting changes in HDX during ligand-binding experiments, we recommend optimizing ligand concentrations and ensuring adequate time for complex formation to ensure maximum protein occupancy. Ligand and protein concentrations used during labeling should be stated, as well as the dissociation constant (if known).

The ligand (NRAS peptide and its variants) was co-purified with our protein at 1:1 stoichiometry. Mass spec and NMR confirm the presence and identity of the bound peptide. The reported IC₅₀ (determined in our manuscript) is 260 nM. HDX-MS experiments were performed in a sufficiently high enough concentration of peptide/MHC-I complex (i.e., above the dissociation constant) where the peptide is expected to be stably bound to the MHC-I. HDX-MS detected bound peptide with very high signal to noise, confirming this.

25.

In comparative HDX-MS analyses, we recommend following good practices in experiment design to control variation. Analysis should be randomized. Our recommendation is to avoid collecting data for all states sequentially or collecting technical replicates sequentially for one state. By doing so, one can mitigate the effects of any drifts in instrument parameters, day-to-day variability (in case data are collected over multiple days), temperature fluctuations or any other parameter that might affect the HDX-MS experiment.

Independent triplicate data collection was completed by different people in a long duration of six months.

Data analysis and data presentation guidelines

21.

When conducting peptide identification using MS/MS, the spectral search database should include the sequence of all major proteins present in the sample and introduced during workup, to prevent false peptide identifications. This is especially advisable for complex sample types. The list should minimally include the proteases used and any major protein contaminants. The composition of the database used for searching should be reported.

All peptide and protein samples used in the study were added to the database used by Thermo Proteome Discoverer v2.4 to identify peptide fragments corresponding to HLA-A*01:01, hβ_{2m}, and various NRAS peptides

22.

Peptide identification criteria should be included in the text, based on the search tool used (that is, scoring cut-off and its statistical basis). The name of the search software (and version) should be provided, with parameters appropriate for the mass spectrometer used.

Thermo Proteome Discoverer v2.4 software was used with standardized parameters.

23.

For quality control purposes, the output of automated HDX-MS computational routines should be supported by an inspection of the raw data, including spectral assignments and isotopologue detection. A summary of the HDX data should be reported in a table (for example, Table 1) with the following information for each protein included within the study: (1) HDX reaction details, for example, pH and temperature; (2) HDX time course, for example, what time points were analyzed; (3) number of peptides analyzed (that is, the total number of peptides for which the deuterium content has been analyzed in each dataset); (4) sequence coverage, expressed as a

percentage of amides covered by the peptides for which deuterium content has been measured (rather than all peptides identified in the non-deuterated experiment); (5) average peptide length and redundancy; (6) a quantitative measure of the repeatability of deuterium measurement, for instance, the average (mean, median, root-mean-square) standard deviation from replicate (technical or biological) measurements of the deuterium content of all or representative peptides from one or more time points; and (7) threshold for significant differences in HDX (a threshold value interpreted as representing a significant difference in HDX between examined protein states based on the quantitative measure of repeatability. We recommend that such a table be provided in the supplementary material for all HDX manuscripts, similar to the convention in the X-ray crystallography field of reporting collection/refinement statistics⁵⁵. We include an example of the HDX summary table as a downloadable template spreadsheet (Supplementary Table 1) to encourage the community to include such data in their reporting, and to do so using a standardized and readily accessible format. We also recommend that a peptide coverage map—a figure showing the identified peptides used to extract HDX information mapped onto the sequence of the protein studied—be included in the manuscript or in the supplementary material.

See Supplementary HDX tables 5-8.

24.

When reporting explicitly on the change of HDX in a peptide owing to, for example, the presence of a binding partner, a peptide uptake plot should be provided, plotting each labeling time with the per-peptide standard deviation. Appropriate statistical analyses should be applied to all reports of differences between states. Furthermore, HDX information from multiple charge states and overlapping peptides should be used to add certainty to any conclusions.

See fig. 4b, 5e, and supplementary fig. 13.

25.

A common data presentation format for HDX-MS data involves color-mapping the time-resolved, peptide-resolution data onto three-dimensional, static, atomic-resolution structures. Inappropriate mapping can result in a loss of information. When mapping HDX data onto structures, scientists should explicitly state their mapping methodology and at which time points data are depicted, and this approach should be based on a quantitative and statistical argument that is applied to the entire dataset. Such an encompassing method avoids selective presentation and permits a balanced assessment of biologically relevant findings. In cases where a difference in HDX is mapped, regions should be carefully indicated for which there is no sequence coverage. Furthermore, regions that exhibit 'no significant difference' should not be interpreted as 'no change'. Rather, it means no detectable difference within the kinetic regime had been detected. The mapping should be accompanied with an explicit statement of this nature. Finally, both authors and readers must recognize that imposing HDX-MS results (measured in a dynamic solution-phase environment) on static structures can easily bias interpretation of the results. Two very different biophysical measurements are being combined, so it is difficult to convey the magnitude of conformational change on a single static structure. Interpretation is most accurate when based on HDX uptake plots.

See fig. 4c, 5f, and supplementary fig. 9, 11, and 12.

Supplemental data presentation recommendations

31.

A supplementary 'HDX data' table corresponding to all the peptides included in the study, including peptides that show no significant difference between states, should be presented. Two examples of such a table are shown in Supplementary Tables 2 and 3; either can be used depending on the output format of the HDX-MS data analysis software in use. We include these downloadable template spreadsheets to encourage the community to include such data in their reporting and to do so using this standardized and readily accessible format. The HDX data table greatly simplifies access to the acquired data for other scientists and will enable any downstream data processing, including use of the data for computational modeling. The minimum requirements are peptide sequence (start and end numbering and sequence), peptide monoisotopic mass (uncharged), chromatographic retention time, mean deuterium uptake (shift in average/centroid mass, without any correction for back-exchange) and the standard deviation, for each labeling time, with a clear indication of the number (n) and nature of replicates (technical/biological) used to determine this value. These data should be provided for all states measured (for example, apo and with a binding partner). If a maximally labeled control and quench exchange control samples are analyzed, similar data should be reported for these as well.

See Supplementary HDX tables 5-8.

32.

Provided that an HDX summary table (Table 1) and an HDX data table (Supplementary Tables 2 and 3) are provided, only deuterium uptake plots for peptides that are explicitly referenced within the main body of the manuscript are necessary to include (in the manuscript or supplementary information). These plots should show the appropriate deuteration values across all measured labeling times, with standard deviations between technical repeats indicated.

See Supplementary HDX tables 5-8.

33.

For peptides undergoing substantial EX1 or mixed exchange profiles⁵⁶, we recommend the inclusion of raw deuterium incorporation mass spectra, or the use of deuterium incorporation plots that show the different bimodal populations (bubble plots can be a useful data representation for this type of data; see ref. 57).

Search engine result files that contain peptide identification (SEARCH), mass spectrometer output files (RAW), and peak list files (PEAK) have been deposited into ProteomeXchange repository¹ with the dataset identifier PXD044838.

Reviewer account details:

Username: reviewer_pxd044838@ebi.ac.uk

Password: LrMO1k6o

REVIEWERS' COMMENTS

Reviewer #1 (Remarks to the Author):

The authors have addressed my concerns and I am supportive of publication.

Reviewer #3 (Remarks to the Author):

We are satisfied with the changes made. The additional analyses of the NOE violations in the MD simulations are insightful.

Reviewer #4 (Remarks to the Author):

The revised manuscript has addressed my concerns appropriately.

Reviewer #5 (Remarks to the Author):

The authors have generally addressed my concerns, and from what I can tell done an admirable job addressing the far more detailed questions of the other reviewers. I still feel that there is a bit of overselling in the results 1) in that much of the work replicates what we already know about p-HLA complexes, and 2) specific details about conformational and dynamic changes upon and after binding are not likely to be any more actionable here than in other system. However I am happy to be proven wrong, so let's see what develops.

Reviewer #6 (Remarks to the Author):

I appreciate that the authors made substantial efforts to comply with the guidelines of the HDX community. Despite this effort, I still have requests in terms of data presentation to facilitate the understanding of the results for both HDX-MS experts and non-experts. Please add a coverage map that shows not only coverage but also average peptide length and redundancy, as these are important parameters to assess the validity/ quality of the HDXMS results. In addition, I would like to see all the peptide uptake plots obtained, for each replicate (what the authors call kinetic graphs). Having this information is essential. (see example in this paper: <https://www.nature.com/articles/s41467-019-09675-z#Sec21> – sfig.3 and sfig6 to sfig.8). At the moment that data is available and presented in the excel sheets but it is unlikely anyone will take the time to replot what the authors have most likely already plotted.

Please find below detailed feedback on the comments made by the previous HDXMS expert.

1. Fig 4c attempts to show a comparative analysis by requiring the reader to compare the “empty” vs NRASQ61K form. Firstly, these should be referred to as the free vs bound samples throughout, and secondly, this is simply too coarse of a comparison to help the reader. Difference plots should be used to support a “delta” deuteration visualization of the datasets.

The authors have only partially addressed this comment. While they have established difference plots, they are not readily available to the reader. Difference plots are typically presented as a ΔD as a function of the amino acid sequence, or as a function of the peptide number. Different options are available for the presentation of this ΔD plot (butterfly plot, differential heat map plot, woods plot – the authors can find nice examples in Kasper Rand papers). I don't have a personal favorite but Fig s9 is as polished and qualitative as fig 4c and does not add much information.

2. To support this difference plot, the authors must establish the statistical criteria they used to make the assessment of difference. A simple visual of the two kinetics datasets is insufficient. I strongly recommend they follow a recently-published “guidelines” article published in Nature Methods 3 years ago (Masson et al., Nat. Methods 2019, 16, 595-602).

The authors have addressed that point.

3. These guidelines also contain information on supplementary datasets that should be provided for the reader to gauge the completeness of the study. Note in particular that the mapping of deuteration changes (or kinetics) should only be based on the peptides that survive deuteration analysis, and not simply the sequence map of the undeuterated data.

In that regard, I encourage the authors to provide all their peptide uptake plots (kinetic graphs) either as supplementary data, or by uploading them on a repository like figshare.

4. Figure 4a shows a peptide that appears to either demonstrate EX1/EXX kinetics (139-154), overlap, or partial binding. It adds concern over the way data analysis was conducted, as this peptide is indicated to demonstrate a large difference in deuteration upon neoepitope binding (Fig 4b). As the guidelines will indicate, a clearer statement on selection criteria may help, in addition to the provision of all kinetics curves if possible.

The authors have addressed that point but again, having all the kinetic curves will help.

5. In connection with this concern, I cannot see what concentration excess of neoepitope to HLA was used in the HDX study. This should be provided as I'm sure it was done to saturate binding (note that upon deuteration there is a high dilution, requiring concentration excess in most cases). And if there was a large excess, how did the authors manage to detect changes in the NRAS peptides themselves? Information is missing.

The authors have addressed that point.

REVIEWERS' COMMENTS

Reviewer #1 (Remarks to the Author):

The authors have addressed my concerns and I am supportive of publication.

We thank the reviewer for their constructive feedback during the review process.

Reviewer #3 (Remarks to the Author):

We are satisfied with the changes made. The additional analyses of the NOE violations in the MD simulations are insightful.

We thank the reviewer for their constructive feedback during the review process.

Reviewer #4 (Remarks to the Author):

The revised manuscript has addressed my concerns appropriately.

We thank the reviewer for their constructive feedback during the review process.

Reviewer #5 (Remarks to the Author):

The authors have generally addressed my concerns, and from what I can tell done an admirable job addressing the far more detailed questions of the other reviewers. I still feel that there is a bit of overselling in the results 1) in that much of the work replicates what we already know about p-HLA complexes, and 2) specific details about conformational and dynamic changes upon and after binding are not likely to be any more actionable here than in other system. However I am happy to be proven wrong, so let's see what develops.

We thank the reviewer for their positive appraisal of our work, and for keeping an open and unbiased perspective on the broader significance of our results.

Reviewer #6 (Remarks to the Author):

I appreciate that the authors made substantial efforts to comply with the guidelines of the HDX community. Despite this effort, I still have requests in terms of data presentation to facilitate the understanding of the results for both HDX-MS experts and non-experts. **Please add a coverage map that shows not only coverage but also average peptide length and redundancy, as these are important parameters to assess the validity/ quality of the HDXMS results. In addition, I would like to see all the peptide uptake plots obtained, for each replicate (what the authors call kinetic graphs). Having this information is essential. (see example in this paper: <https://www.nature.com/articles/s41467-019-09675-z#Sec21> – sfig.3 and sfig6 to sfig.8).** At the moment that data is available and presented in the

excel sheets but it is unlikely anyone will take the time to replot what the authors have most likely already plotted.

We thank the reviewer for pointing out the importance of including sequence coverage maps and peptide uptake plots for better visualization of the data set. The sequence coverage maps, sequences, and peptide uptake plots are now included. The sequence coverage maps present sequences of peptide-free, NRAS^{Q61K}, NRAS^{Q61}, NRAS^{Q61H}, NRAS^{Q61L}, and NRAS^{Q61R} peptide-loaded HLA-A*01:01/hβ₂m in a single chain format, heavy chain followed by hβ₂m and peptide, connected by G linkers. G linkers are not included in the actual protein sequences. Peptide fragments from replicate 1, 2, and 3 are colored in blue, green, and red, respectively. The sequence coverage maps and the protein sequences can be accessed from Figshare: <https://figshare.com/s/5352674efdb3865a4e46>.

Please find below detailed feedback on the comments made by the previous HDXMS expert.

1. Fig 4c attempts to show a comparative analysis by requiring the reader to compare the “empty” vs NRASQ61K form. Firstly, these should be referred to as the free vs bound samples throughout, and secondly, this is simply too coarse of a comparison to help the reader. Difference plots should be used to support a “delta” deuteration visualization of the datasets.

The authors have only partially addressed this comment. While they have established difference plots, they are not readily available to the reader. Difference plots are typically presented as a ΔD as a function of the amino acid sequence, or as a function of the peptide number. Different options are available for the presentation of this ΔD plot (butterfly plot, differential heat map plot, woods plot – the authors can find nice examples in Kasper Rand papers). I don't have a personal favorite but Fig s9 is as polished and qualitative as fig 4c and does not add much information.

Based on the reviewer's comment and the exemplary Kasper Rand papers, we have modified the presentation of the ΔD plot between peptide-free and NRASQ61K-peptide-loaded HLA-A*01:01 at 600 sec by adding a butterfly plot. The determined threshold of significant differences in delta HDX (0.297D, Average+1SD) is labeled in the updated Supplementary Figure 9, and the original raw data numbers can be traced in supplementary HDX table_8.

2. To support this difference plot, the authors must establish the statistical criteria they used to make the assessment of difference. A simple visual of the two kinetics datasets is insufficient. I strongly recommend they follow a recently-published “guidelines” article published in Nature Methods 3 years ago (Masson et al., Nat. Methods 2019, 16, 595-602).

The authors have addressed that point.

3. These guidelines also contain information on supplementary datasets that should be provided for the reader to gauge the completeness of the study. Note in particular that the mapping of deuteration changes (or kinetics) should only be based on the peptides that survive deuteration analysis, and not simply the sequence map of the undeuterated data.

In that regard, I encourage the authors to provide all their peptide uptake plots (kinetic graphs) either as supplementary data, or by uploading them on a repository like figshare.

The %D uptake plots are included as the supplementary data and uploaded into Figshare:

<https://figshare.com/s/801ba9327b7ea8af11d6>
<https://figshare.com/s/0df657428ee6fb685d0d>
<https://figshare.com/s/7da19991e6c60edcdc10>
<https://figshare.com/s/7c2f86d782ac1b47ae7e>
<https://figshare.com/s/5eebe077c56f8ac107eb>
<https://figshare.com/s/03fdea084b4a8ca6b26e>

4. Figure 4a shows a peptide that appears to either demonstrate EX1/EXX kinetics (139-154), overlap, or partial binding. It adds concern over the way data analysis was conducted, as this peptide is indicated to demonstrate a large difference in deuteration upon neoepitope binding (Fig 4b). As the guidelines will indicate, a clearer statement on selection criteria may help, in addition to the provision of all kinetics curves if possible.

The authors have addressed that point but again, having all the kinetic curves will help.

The kinetics plots are included as the supplementary data and uploaded into Figshare:

<https://figshare.com/s/801ba9327b7ea8af11d6>
<https://figshare.com/s/0df657428ee6fb685d0d>
<https://figshare.com/s/7da19991e6c60edcdc10>
<https://figshare.com/s/7c2f86d782ac1b47ae7e>
<https://figshare.com/s/5eebe077c56f8ac107eb>
<https://figshare.com/s/03fdea084b4a8ca6b26e>

5. In connection with this concern, I cannot see what concentration excess of neoepitope to HLA was used in the HDX study. This should be provided as I'm sure it was done to saturate binding (note that upon deuteration there is a high dilution, requiring concentration excess in most cases). And if there was a large excess, how did the authors manage to detect changes in the NRAS peptides themselves? Information is missing.

The authors have addressed that point.